# Novel Amperometric Sensor Based on Glassy Graphene for Flow Injection Analysis

**DOI:** 10.3390/s25082454

**Published:** 2025-04-13

**Authors:** Ramtin Eghbal Shabgahi, Alexander Minkow, Michael Wild, Dietmar Kissinger, Alberto Pasquarelli

**Affiliations:** 1Institute of Electronic Devices and Circuits, Ulm University, 89069 Ulm, Germany; dietmar.kissinger@uni-ulm.de; 2Institute for Quantum Optics, Ulm University, 89069 Ulm, Germany; 3Institute of Functional Nanosystems, Ulm University, 89069 Ulm, Germany; alexander.minkow@uni-ulm.de; 4Diamond Materials GmbH, 79108 Freiburg, Germany; michael.wild@diamond-materials.de

**Keywords:** glassy graphene, pyrolyzed photoresist film, polycrystalline diamond, rapid thermal annealing, adrenaline

## Abstract

Flow injection analysis (FIA) is widely used in drug screening, neurotransmitter detection, and water analysis. In this study, we investigated the electrochemical sensing performance of glassy graphene electrodes derived from pyrolyzed positive photoresist films (PPFs) via rapid thermal annealing (RTA) on SiO_2_/Si and polycrystalline diamond (PCD). Glassy graphene films fabricated at 800, 900, and 950 °C were characterized using Raman spectroscopy, scanning electron microscopy (SEM), and atomic force microscopy (AFM) to assess their structural and morphological properties. Electrochemical characterization in phosphate-buffered saline (PBS, pH 7.4) revealed that annealing temperature and substrate type influence the potential window and double-layer capacitance. The voltammetric response of glassy graphene electrodes was further evaluated using the surface-insensitive [Ru(NH_3_)_6_]^3+/2+^ redox marker, the surface-sensitive [Fe(CN)_6_]^3−/4−^ redox couple, and adrenaline, demonstrating that electron transfer efficiency is governed by annealing temperature and substrate-induced microstructural changes. FIA with amperometric detection showed a linear electrochemical response to adrenaline in the 3–300 µM range, achieving a low detection limit of 1.05 µM and a high sensitivity of 1.02 µA cm^−2^/µM. These findings highlight the potential of glassy graphene as a cost-effective alternative for advanced electrochemical sensors, particularly in biomolecule detection and analytical applications.

## 1. Introduction

Carbon-based electrodes, including graphite, glassy carbon (GC), and boron-doped diamond (BDD), are widely employed in analytical and industrial electrochemistry due to their biocompatibility, chemical stability, and, in some cases, low background currents, which contribute to improved detection limits [1]. BDD, entirely composed of sp^3^-hybridized carbon, stands out for its exceptional electrochemical properties, including a wide potential window, minimal background currents, and remarkable resistance to electrochemical fouling [2,3]. These attributes make BDD highly desirable for demanding electroanalytical applications. However, its widespread adoption is hindered by high production costs and scalability limitations. To address these challenges, there has been growing interest in the development of low-cost, disposable carbon-based electrodes that are compatible with lithographic techniques, particularly for electrochemical sensor applications.

Pyrolyzed photoresist films (PPFs) have emerged as a promising alternative due to their high performance, cost-effectiveness, and simple fabrication process, making them well suited for microelectrodes in electroanalytical applications such as neurotransmitter detection [4,5,6,7]. PPFs are typically fabricated by spin-coating photoresists onto substrates such as silicon, followed by pyrolysis at ≈1000 °C under an inert [8] or reducing atmosphere [9]. Structurally, PPFs share similarities with GC, consisting of graphitic domains embedded in an amorphous carbon matrix. However, they offer distinct advantages, including atomically smooth surfaces (RMS roughness < 0.5 nm) and lower capacitive currents, which enhance their electrochemical performance in specific applications [7,10,11,12].

The properties of PPFs derived from photoresists are influenced by both the type of precursor material (positive or negative photoresist) and the pyrolysis temperature [13,14]. Studies have shown that positive photoresists yield higher conductivity than negative photoresists [6], exhibit shrinkage of approximately 80%, and have capacitance values ranging from 7 to 100 µF/cm^2^ [11]. Previous studies [11,12,15] have demonstrated that increasing the annealing temperature improves graphitization, thereby reducing resistivity and enhancing electron transfer kinetics for redox couples such as [Ru(NH_3_)_6_]^3+/2+^ and [Fe(CN)_6_]^3−/4−^. Despite these advantages, PPF electrodes have shown lower efficiency in detecting catecholamine neurotransmitters compared to GC, mainly due to weaker analyte adsorption [7].

Catecholamine neurotransmitters such as adrenaline (epinephrine, EP) and dopamine (DA) play critical roles in physiological processes, including cardiovascular regulation and neurological function [16]. Abnormal levels of these neurotransmitters are associated with various medical conditions, such as neurodegenerative diseases, cardiovascular disorders, and hormonal imbalances [17]. Consequently, developing simple, reliable, and sensitive analytical methods for their detection in different matrices is essential. Given their electroactive nature, electrochemical methods provide a direct and sensitive approach for neurotransmitter detection. However, electrode material properties, including surface chemistry, conductivity, and adsorption behavior, significantly influence the detection efficiency. Del Campo et al. [12] reported that DA detection on PPFs is less effective than that on GC due to weaker adsorption. While electrochemical oxidation can modify the electrode surface by introducing hydroxyl functional groups, enhancing DA physisorption, this process also increases the background current, which can negatively affect the detection limits.

Graphene has emerged as a promising alternative electrode material due to its high surface area, excellent conductivity, and strong analyte adsorption properties, which improve the detection sensitivity [18]. The International Union for Pure and Applied Chemistry (IUPAC) defines graphene as “a single carbon layer of graphite structure, describing its nature by analogy to a polycyclic aromatic hydrocarbon of quasi-infinite size” [19].

However, in practical applications, graphene often exists in various forms, including single-layer, few-layer (2–9 layers), and multilayer graphene (≥10 layers). This broad range of thicknesses deviates from the strict IUPAC definition, yet remains distinct from graphite, which requires long-range stacking of graphene layers along the c-axis in an ordered ABA or ABC configuration, a criterion not always met by multilayer graphene [20].

Graphene has gained significant attention due to its remarkable properties, including high mechanical strength, exceptional electrical conductivity, large surface area, flexibility, tunable surface chemistry, and biocompatibility [21,22]. These characteristics make it a highly promising material for various electrochemical applications, particularly in biosensing [23], where it has been explored for detecting targets such as proteins [24], DA [25], and nucleic acid [26]. One of the most common fabrication methods for graphene-based electrodes involves drop-casting a graphene dispersion onto commercial electrode surfaces, such as GC electrodes [27]. However, this approach has notable drawbacks. First, graphene sheets tend to agglomerate due to van der Waals interactions and π-π stacking, which reduce their effective surface area and electrochemical activity. Second, the weak adhesion between the graphene layer and the underlying substrate, also caused by van der Waals interactions, compromises the electrode’s long-term stability, especially under harsh operating conditions [28]. To overcome poor interfacial adhesion, direct fabrication of graphene-on-diamond heterostructures has emerged as a promising alternative [29].

Unlike BDD electrodes, sp^2^-carbon-based electrodes are more susceptible to fouling and require frequent regeneration through ultrasonic cleaning, which compromises their long-term stability [30]. Consequently, conventional drop-casting methods for transferring graphene onto diamond are not ideal due to the weak interfacial bonding between the materials. Instead, the direct synthesis of graphene on diamond provides a more robust and stable electrode platform.

Several studies have explored different strategies for integrating graphene with diamond. The earliest significant work was reported in 2011 by García et al. [31], who demonstrated the growth of multilayer graphene on diamond via carbon precipitation during the cooling phase of a high-temperature annealing process with a nickel catalyst. Building on this approach, in 2016, Berman et al. [32] developed a direct transformation method, converting ultrananocrystalline diamond into high-quality graphene layers through rapid thermal annealing (RTA) using a nickel thin film as a catalyst. They found that lower annealing temperatures produced predominantly monolayer graphene, while increasing the temperature to 1000 °C led to multilayer formation. Subsequent studies expanded the application of graphene-on-diamond electrodes.

In 2018, Yuan et al. [33] fabricated graphene–diamond hybrid electrodes using high-pressure high-temperature (HPHT) diamond as a carbon source and copper as a catalyst at 1100 °C in a tube furnace with an 8 sccm H_2_ flow. Their electrode exhibited excellent electrochemical performance for DA sensing, with a linear response range of 5 μM to 2 mM and a low detection limit of 200 nM using differential pulse voltammetry (DPV). That same year, Marcu et al. [34] demonstrated that depositing a graphene suspension (graphene, isopropyl alcohol, and Nafion) onto BDD electrodes significantly improved the sensitivity, reaching 3.00 µA cm^−2^/µM within the 1 to 10 µM range using linear sweep voltammetry (LSV).

In 2020, Pei et al. [35] introduced a graphene-functionalized self-supported boron-doped diamond (G/SBDD) electrode using an in situ graphene modification technique for the electrochemical detection of trace Pb^2+^ in seawater. The most recent advancements have focused on optimizing graphene growth on boron-doped diamond by controlling the doping levels. In 2024, Liu et al. [36] investigated how boron doping in diamond substrates influences graphene growth through catalytic thermal treatment. Their findings revealed that heavily doped diamond substrates promote the formation of high-quality, few-layer graphene with extensive surface coverage. Despite these developments, research on graphene growth specifically on polycrystalline diamond (PCD) remains limited. The only notable study in this area is the 2023 work by Che et al. [37], which examined the in situ growth mechanism and characteristics of graphene catalyzed by a nickel coating.

Recently, a new class of carbon-based material known as glassy graphene (GG) has emerged as a promising alternative electrode material. Structurally, glassy graphene occupies an intermediate state between glassy carbon and graphene, inheriting beneficial properties from both. It is more crystalline than glassy carbon yet retains a distorted lattice compared to graphene. This unique structure grants glassy graphene excellent chemical and thermal stability, comparable to glassy carbon, while also maintaining high conductivity and flexibility similar to graphene, making it highly attractive for electrochemical applications. Dai et al. [38] successfully fabricated ultrasmooth GG thin films on quartz at 850 °C using glucose and polyethylene imine as precursors through an annealing process. They demonstrated that GG thin films exhibit conductivity, transparency, and flexibility comparable to graphene, while also possessing the mechanical and chemical stability characteristic of GC. To date, the electrochemical performance of GG remains largely unexplored, presenting an open opportunity for further investigation.

Since the electrochemical detection method influences the sensitivity and limit of detection (LOD) of the target analyte, flow injection analysis (FIA) presents several advantages. These include cost-effectiveness, automation, compatibility with various detection techniques, minimal reagent consumption, continuous monitoring, and rapid analysis [39,40]. Integrating sp^2^-carbon-based electrodes into FIA presents a key advantage: reduced surface contamination. In contrast to stationary electrochemical cells, FIA limits the sample’s contact time with the electrode, minimizing surface fouling by reaction products. Additionally, coupling FIA with amperometric detection improves sensitivity and enhances the signal-to-noise ratio (S/N) compared to diffusion-limited voltammetric techniques [40].

In this study, we present a novel approach for fabricating GG from an image reversal photoresist (AZ 5214E) using a nickel catalyst and RTA on PCD and SiO_2_/Si substrates. This work not only demonstrates the feasibility of GG synthesis via RTA but also highlights, for the first time, the significant influence of substrate type on surface microstructure and electrochemical performance. To assess the electrochemical properties of GG electrodes, we employed cyclic voltammetry (CV), electrochemical impedance spectroscopy (EIS), and FIA with amperometric detection of adrenaline. CV provides insight into charge transfer kinetics, while EIS evaluates interfacial properties, offering a deeper understanding of electrode behavior. To our knowledge, this is the first study to investigate the integration of GG with FIA for electrochemical sensing, paving the way for future advancements in high-performance carbon-based electrode materials.

## 2. Materials and Methods

### 2.1. Materials and Reagents

All chemicals were used as received without further purification, and all solutions were prepared using ultrapure water (resistivity ≥ 18 MΩ·cm). Potassium hexacyanoferrate (III) was obtained from Merck KGaA (Darmstadt, Germany), while potassium nitrate was sourced from VWR Chemicals. Epinephrine and hexaammineruthenium (II) chloride were supplied by Sigma-Aldrich (Germany). Prior to use, solutions were deoxygenated by bubbling pre-purified nitrogen gas for a minimum of 30 min. A 30 mM adrenaline stock solution was prepared by dissolving the required mass of adrenaline powder in 50 mL of 32% hydrochloric acid (Merck), followed by dilution with phosphate-buffered saline (PBS, pH 7.4, VWR Chemicals) to obtain working concentrations of 3 µM, 10 µM, 30 µM, 100 µM, 300 µM, and 1 mM. An image reversal photoresist (AZ 5214E) and developer (AZ 726 MIF), along with acetone and isopropanol, were sourced from MicroChemicals GmbH (Ulm, Germany).

Silicon wafers and PCD substrates were cleaned using freshly prepared piranha solution (H_2_SO_4_:H_2_O_2_, 2:1). The sulfuric acid (97%) and hydrogen peroxide (30%) were obtained from MicroChemicals GmbH. Silicon wafers used in this study were prime-grade, float-zone (FZ) silicon with a resistivity of 5000–1,000,000 Ω·cm, (100) orientation, 2-inch diameter, and 300 μm thickness, purchased from MicroChemicals GmbH.

### 2.2. Material Characterization and Electrochemical Measurements

The surface morphology of diamond was analyzed using a Zeiss LEO 1550 scanning electron microscope (SEM) at an acceleration voltage of 15 kV, while pyrolyzed photoresist films and GG films were examined at 4 kV. Surface roughness measurements were conducted using a Veeco Dimension 3100 atomic force microscope (AFM) in tapping mode utilizing an n-doped silicon tip (0.01–0.02 Ω·cm) for high-resolution imaging. Raman spectroscopy was carried out with a Renishaw inVia confocal Raman system, equipped with a Leica DM 2500 microscope. Spectra were acquired using a 633 nm laser, which effectively reduces fluorescence interference and is useful for analyzing structural defects [32,41]. The laser power was set to 4 mW, corresponding to 5% of the instrument’s maximum power, with an acquisition time of 10 s.

For electrochemical measurements, GC derived from PPFs and GG chips were mounted onto custom carrier boards (Roger 4003) using a flip-chip technique. Silver epoxy glue (EC 201 Polytec PT, Waldbronn, Germany) was applied to the bonding pads to establish electrical contact, while non-conducting, biocompatible epoxy (Polytec EP 653, Waldbronn, Germany) was used to seal all gaps, ensuring mechanical stability, tightness, and minimal electrochemical noise. The carrier boards were designed with a slotted opening to expose the GC and GG chips for sensing, with the reference electrode (RE) positioned on the left and the counter electrode (CE) on the right. The fabrication of RE and CE has been previously described [42].

Cyclic voltammetry (CV) measurements were conducted using a custom-built electronic front-end controlled via a LabVIEW-based program. The system includes a four-channel potentiostat with integrated transimpedance amplifiers and a differential signal conditioning stage, enabling precise electrode biasing and high-resolution signal acquisition. This setup allows flexible biasing protocols for CV and chronoamperometry. For a detailed description of the instrumentation and signal processing, refer to [42].

The initial CV application focused on evaluating the double-layer capacitance (C_dl_) of PPFs and GG electrodes in PBS (pH 7.4), closely matching the physiological pH of 7.365. A triangular wave potential scan was applied from −0.3 V to 0.3 V at a scan rate of 100 mV s^−1^. The C_dl_ was calculated from the CV data using the following integral equation:(1)$$C_{dl}=\frac{1}{A\upsilon (E_{2}-E_{1})}\int_{E_{1}}^{E_{2}}i\left(E\right)dE$$

Here, *E_1_* and *E_2_* correspond to the lower and upper potential limits, respectively, while *i(E)* represents the recorded current at each potential during the CV scan. The scan rate is denoted by *υ*, while *A* refers to the electrode area participating in the measurement.

Electrochemical impedance spectroscopy (EIS) measurements were performed using a PARSTAT 273 potentiostat (Princeton Applied Research, Oak Ridge, TN, USA) with GG chips mounted on carrier boards in a three-electrode configuration. The impedance response was recorded at the formal redox potential of 1 mM [Ru(NH_3_)_6_]^3+/2+^ in 0.1 M KNO_3_, as determined from CV measurements. A frequency range of 0.1 Hz to 10 kHz was applied with a 10 mV AC perturbation. The resulting impedance spectra were analyzed and fitted using ZsimpWin software (version 2.00).

Flow injection analysis (FIA) was performed using a custom-designed microfluidic flow cell, integrating CV for preliminary scans followed by chronoamperometric detection. The FIA system comprised an eight-channel perfusion stand with software-controlled electromagnetic pinch valves. Six channels were employed for delivering adrenaline solutions of varying concentrations (3 μM, 10 μM, 30 μM, 100 μM, 300 μM, and 1000 μM) prepared in PBS, while a separate channel was dedicated to PBS rinsing.

A 12-roller peristaltic pump (Ismatec, Wertheim, Germany) maintained a steady flow rate. The perfusion system was controlled via a LabVIEW-integrated function, as well as the real-time data acquisition. To initiate FIA amperometric detection, the microfluidic chamber was first filled with PBS. Adrenaline at 3 μM was then injected for 192 s at a flow rate of 0.62 mL/min, covering eight CV scan cycles from 0 V to 1.2 V at 100 mV s^−1^ to assess potential electrode fouling. This corresponded to an approximate sample volume of 1.98 mL per concentration.

After perfusion, the current response was recorded for an additional 192 s under stationary conditions of diffusion, after which the system was rinsed with PBS for 192 s. This sequence was systematically repeated for each subsequent adrenaline concentration.

### 2.3. Fabrication of Pyrolyzed Photoresist Microelectrodes

The fabrication of pyrolyzed photoresist microelectrodes was carried out on both silicon wafers and PCD substrates. For Si-based substrates, the wafers were first cleaned with piranha solution to remove organic contaminants, followed by the deposition of a 1 µm thick SiO_2_ insulating layer via plasma-enhanced chemical vapor deposition (PECVD, Plasmalab 80Plus) to ensure electrical isolation. The SiO_2_-coated wafers were then sequentially cleaned by sonication in acetone and isopropanol. For PCD substrates, 5 × 10 mm^2^ polycrystalline diamond samples, grown using an ellipsoidal microwave-powered plasma reactor [42], were treated in chromosulfuric acid at 80 °C for 20 min to eliminate residual nondiamond carbon (NDC) impurities.

An additional piranha solution cleaning step was performed to ensure a pristine surface. To define the microelectrode patterns, an image reversal photoresist (AZ 5214E) was dispensed by spin-coating onto both the SiO_2_/Si and PCD substrates at 4000 rpm for 60 s, followed by a soft bake at 110 °C for 90 s under ambient conditions.

The samples were then aligned with a photolithographic mask and exposed to UV light (405 nm, 4 mW cm^−2^) for 20 s using an MJB4 mask aligner (SUSS MicroTech SE, Graching, Germany).

To exploit the negative-tone characteristics of the resist, the samples underwent a reversing post-exposure bake at 110 °C for 90 s, followed by a flood UV exposure for 40 s. Finally, the photoresist patterns were developed by immersion in AZ 726 MIF developer, revealing the desired electrode structures for the subsequent pyrolysis process.

The pyrolysis process was performed using a Steag AST SHS 2000 rapid thermal annealing (RTA) system under a high-purity nitrogen (N_2_, 3000 sccm) atmosphere. To remove residual air, the chamber was purged three times with nitrogen and evacuated before processing. Since significant weight loss occurs between 250 °C and 500 °C [11], which can compromise the integrity of the microelectrode geometry, a two-step pyrolysis process was implemented. In the first step (curing step), the temperature was ramped to 350 °C over 10 min and held for 60 min under N_2_ to stabilize the structure while preventing excessive material loss.

The samples were then allowed to cool down to room temperature over 10 min. In the second step, designed to complete the graphitization process, the temperature was increased to 450 °C at a rate of 7.5 °C/s and held for 5 min before being further ramped up to 950 °C at 20 °C/s, where it was maintained for 1 min. The samples were then allowed to cool down to room temperature under a continuous nitrogen flow of 5000 sccm. Following pyrolysis, the SiO_2_/Si substrates were cut into 5 × 10 mm^2^ pieces, and both the PPF/SiO_2_/Si and PPF/PCD samples were analyzed using surface and electrochemical characterization techniques. The vertical reduction in film thickness due to pyrolysis was measured using a Dektak profilometer (Bruker, Billerica, MA, USA.).

### 2.4. Fabrication of Glassy Graphene (GG) Microelectrodes

The synthesis of arrays with four GG microelectrodes was carried out on both SiO_2_/Si and PCD substrates. The process began with the patterning of pyrolyzed photoresist microelectrodes using an image reversal photoresist (AZ 5214E) and UV photolithography.

Following the patterning, a thin layer (50 nm) of nickel (Ni) was deposited onto the developed pyrolyzed photoresist microelectrodes via thermal evaporation using a Pfeiffer PLS 570 system. To selectively retain the Ni coating only on the microelectrodes, a lift-off process was employed. This involved sonication in acetone for 10 min to dissolve the remaining photoresist and remove excess Ni, followed by rinsing in isopropanol to ensure a clean surface. The resulting Ni-coated pyrolyzed photoresist microelectrodes were then ready for the final transformation into GG.

The conversion of Ni-coated PPF microelectrodes into GG was performed using RTA under the same protocol as the second step of the pyrolysis process. After annealing, the samples were used directly for surface and electrochemical characterization without any additional treatment. Figure 1 presents an SEM image of a diced GG chip with four stripe electrodes alongside its integration onto a carrier board.

Figure 2 provides a detailed overview of the experimental workflow, outlining the key fabrication steps and characterization techniques. The process begins with the microfabrication of AZ 5214E photoresist patterns using photolithography, which defines the electrode geometry. This is followed by the fabrication of PPFs through RTA under an N_2_ atmosphere. Subsequently, a thin layer of nickel is thermally evaporated onto the PPF electrodes, serving as a catalyst for the transformation into GG via a second RTA process. The resulting electrodes are then subjected to structural, surface, and electrochemical characterization to evaluate their morphology, composition, and electrochemical performance.

## 3. Results and Discussion

### 3.1. Raman Spectroscopy

#### 3.1.1. Glassy Carbon Derived from AZ 5214E

Raman spectroscopy was employed to investigate the transformation of AZ 5214E films into GC and their subsequent conversion into GG using a thin nickel film as a catalyst. This analysis was conducted on both SiO_2_/Si and polished PCD substrates using the RTA method, providing insights into the structural evolution and substrate-dependent effects on graphene formation. To ensure an accurate peak assignment and a clearer interpretation of the structural features, the Raman spectra for both substrates were baseline-corrected and normalized to their maximum peak intensity, with peaks fitted using a Lorentzian function. Figure 3A presents the Raman spectrum of the PPF at 950 °C for 1 min on a SiO_2_/Si substrate. The spectra reveal two prominent peaks at approximately 1343 cm^−1^ and 1593 cm^−1^, which are characteristic of GC, confirming the successful conversion of the AZ 5214E film into a disordered carbon structure [43,44]. The peak at 1593 cm^−1^, known as the G band, is attributed to the in-plane vibrational modes of sp^2^-bonded carbon atoms, a feature also observed in graphite. This band is indicative of graphitic domains and serves as a marker of sp^2^ carbon ordering within the material [1,45]. In contrast, the peak at 1343 cm^−1^, referred to as the D band, corresponds to the breathing modes of sp^2^ carbon atoms in aromatic rings and is indicative of structural defects or disorder within the carbon lattice [46].

The intensity ratio of these bands (I_D_/I_G_) serves as a crucial metric for evaluating the degree of disorder and the domain size of graphitic regions within the GC structure [47]. A higher I_D_/I_G_ ratio indicates greater structural disorder and defect density, whereas a lower ratio reflects enhanced graphitic ordering and larger sp^2^ domains. Notably, the I_D_/I_G_ ratio observed in this study is lower than previously reported values for PPFs fabricated using conventional heating methods at 1100 °C for 60 min [11]. This suggests that RTA-derived films exhibit a higher degree of sp^2^ carbon ordering with fewer structural defects, likely due to the rapid thermal processing facilitating improved graphitic domain development.

A comparable Raman spectrum is obtained for the PPFs annealed under identical conditions on a diamond substrate (Figure 3B). However, notable differences emerge when compared to the SiO_2_/Si substrate. Specifically, the I_D_/I_G_ ratio is higher on the diamond substrate, and the full width at half maximum (FWHM) of the D band is broader (Table 1). The increased I_D_/I_G_ ratio and D-band broadening suggest that the diamond phonon peak at 1333 cm^−1^ overlaps with the D band, contributing to an apparent increase in disorder in the Raman response. Despite this, the findings highlight the effectiveness of RTA in promoting carbon atom reorganization into more ordered graphitic domains within a short high-temperature processing window, achieving a level of structural refinement beyond that of conventional heating methods.

#### 3.1.2. Glassy Graphene Formation

Raman characterization was performed on nickel-coated glassy carbon films annealed at 700 °C, 800 °C, 900 °C, and 950 °C on a SiO_2_/Si substrate for 1 min via RTA to investigate the nickel-catalyzed transformation into graphene. The intensity ratios (I_D_/I_G_, I_2D_/I_G_) along with the FWHM of the 2D band for the annealed films are summarized in Table 2. The Raman spectrum of the film annealed at 700 °C (Figure 4A) closely resembles that of the uncoated GC derived from pyrolyzed photoresist film (Figure 3A), indicating that graphene formation has not occurred at this temperature. The spectrum features broad peaks at ~1343 cm^−1^ (D band) and ~1593 cm^−1^ (G band) with no detectable 2D band (~2670 cm^−1^), which is a key signature of graphene [45,46]. Additionally, the I_D_/I_G_ ratio is higher compared to the uncoated GC, signifying increased structural disorder within the carbon matrix (Table 2). The absence of graphene formation at 700 °C is likely due to insufficient carbon diffusion into the nickel catalyst at this temperature, preventing the nickel-catalyzed transformation into graphene. Further details on this mechanism will be discussed in the following sections.

In contrast, the Raman spectra of the films annealed at 800 °C, 900 °C, and 950 °C (Figure 4D–L) show a prominent additional feature: a single Lorentzian peak corresponding to the 2D band at ~2670 cm^−1^. The FWHM of this peak ranges from 49.82 to 68.82 cm^−1^ across different locations within each sample. This distinct 2D band is a hallmark of graphene formation and aligns well with Raman spectra commonly reported for graphene synthesized via chemical vapor deposition (CVD) [48,49]. Notably, our findings reveal that sp^2^ carbon atoms within the GC structure can diffuse into the 50 nm nickel film during RTA, facilitating the nickel-catalyzed transformation into graphene. This observation challenges earlier reports suggesting that GC, due to its inherently stable and cross-linked structure, lacks sufficient mobility to diffuse into nickel at 800 °C [31].

The intensity ratio of the 2D band to the G band (I_2D_/I_G_) serves as a key metric for assessing the layer structure of graphene in these films. Variations in the I_2D_/I_G_ ratio across different locations suggest the coexistence of bilayer and multilayer graphene, highlighting the nonuniformity caused by the polycrystalline nature of the nickel catalyst film [50,51]. According to the literature, an I_2D_/I_G_ ratio of 0.8–1.4, along with an FWHM of 45–60 cm^−1^, is indicative of bilayer graphene, whereas values greater than 1.4 are associated with monolayer graphene. Ratios below 0.8 signify the presence of three or more graphene layers [52,53].

Our analysis shows that the film annealed at 800 °C exhibits a higher I_2D_/I_G_ ratio than the film annealed at 950 °C, as reported in Table 2. This trend indicates that increasing the annealing temperature facilitates the growth of additional graphene layers, transitioning from bilayer to multilayer graphene at higher temperatures. This observation was consistently reproduced across multiple samples annealed under the same conditions. These findings align with previous studies on graphene thin films synthesized via RTA using Ni/Cu catalytic layers [54].

Additionally, the I_D_/I_G_ ratio decreases from 0.21 to 0.14 as the annealing temperature increases from 800 °C to 950 °C. This reduction reflects a decline in structural defects, suggesting improved carbon lattice ordering, reduced defect density, and the formation of graphene with larger domain sizes at elevated temperatures. This trend is consistent with prior research [54,55,56,57]. Despite this structural improvement, the persistent D band in the Raman spectra is attributed to ripple structures observed on the nickel catalytic surface, as confirmed by SEM and AFM analyses (see below) [49,52,58]. These ripples induce localized lattice distortions, contributing to defect-related Raman features and reinforcing the impact of surface morphology on graphene quality. Furthermore, the relatively low defect density in the RTA-derived films suggests that the material is best described as GG, a few-layer graphene structure, rather than a perfectly ordered, defect-free graphene lattice.

The Raman spectrum of nickel-coated glassy carbon films annealed at 950 °C for 1 min on PCD (with a representative spectrum shown in Figure 5) closely resembles that of films grown on the SiO_2_/Si substrate, confirming a consistent graphene growth mechanism.

The I_2D_/I_G_ ratio varies between 0.71 and 0.84 across different spots, indicating the formation of multilayer graphene (more than three layers). However, the nonuniformity in the number of graphene layers is more pronounced on PCD than on SiO_2_/Si, highlighting the influence of the diamond substrate. Previous studies suggest that multilayer graphene tends to form at grain boundaries and defects, while monolayer and bilayer graphene preferentially grow on flat diamond facets [37]. In our study, despite the nearly identical D-band FWHM values on PCD (33.81 ± 1.58 cm^−1^) and SiO_2_/Si (33.82 ± 0.47 cm^−1^), the higher I_D_/I_G_ ratio on PCD indicates increased structural disorder (Table 2). Notably, the increased D-band intensity on PCD, without a corresponding broadening of its FWHM, suggests that the diamond grain boundaries influence the structural evolution of the Ni film during annealing. This results in inhomogeneous graphene nucleation and localized defects, distinguishing these grain-boundary-induced defects from the phonon-related broadening typically observed in GC formation on PCD. SEM imaging and EDX mapping (see below) further support this interpretation, revealing localized Ni redistribution at grain boundaries, which likely contributes to the observed Raman features. These findings reinforce the viability of RTA for synthesizing graphene on diverse substrates, including PCD, while highlighting substrate-dependent variations in structural ordering and defect density.

### 3.2. AFM and SEM Characterization

#### 3.2.1. Glassy Carbon Films Derived from AZ 5214E

The surface roughness and morphology of the prepared GC and GG films were analyzed using atomic force microscopy (AFM) and scanning electron microscopy (SEM). The SEM images of the GC films fabricated from AZ 5214E, as shown in Figure 6A, reveal the development of a continuous, smooth film with no visible pores or defects, suggesting excellent structural integrity. Complementary AFM analysis (Figure 6B) confirms this, with a root mean square (RMS) roughness of 0.82 nm, highlighting a highly uniform and potentially dense surface with minimal topographical variation. These findings indicate the successful fabrication of GC films with a well-controlled surface morphology. Profilometer measurements indicate a significant thickness reduction during the RTA process, from approximately 1.3 µm to 160 ± 10 nm. This reduction is attributed to the release of volatile compounds such as H_2_O, CO, and CO_2_ during pyrolysis [11]. Despite this drastic reduction in thickness, the resulting film remains dense and free of detectable porosity, highlighting the annealing process’s effectiveness in compacting the photoresist into a robust GC film.

#### 3.2.2. Glassy Graphene on SiO_2_/Si

Figure 7 presents SEM images of nickel-coated glassy carbon films annealed via RTA at 700 °C. The surface morphology is dominated by the Ni film, with only small carbonaceous islands visible. Raman spectroscopy confirms that these islands retain the spectral characteristics of GC rather than a fully formed GG structure (Figure 4A–C). This suggests that at this temperature, the thermal energy is insufficient to facilitate complete carbon diffusion through the Ni layer or to drive a conventional graphene formation, namely the dissolution–precipitation mechanism of graphene growth in chemical vapor deposition (CVD), where carbon dissolves into Ni during heating and segregates to the surface upon cooling below the solid solubility limit [49,51,59]. However, this process does not adequately explain the lack of graphene formation at 700 °C in the RTA system. Instead, studies [52,60], including those by Sanenger et al. [61], propose an alternative mechanism for RTA-derived graphene growth: metal-induced crystallization (MIC) with layer exchange. In this process, carbon atoms diffuse rapidly through the Ni film during heating, bypassing the dissolution step observed in conventional CVD, and crystallize into graphene on the Ni surface. At 700 °C, the available thermal energy is likely below the threshold required for efficient carbon diffusion, and the limited catalytic activity of nickel prevents carbon atoms from undergoing complete structural reorganization into a graphene lattice [62,63]. As a result, the film remains predominantly disordered, exhibiting characteristics of GC rather than a well-ordered graphene structure. The absence of a distinct 2D band in the Raman spectra further supports this observation.

Unlike the film microstructure observed at 700 °C, nickel-coated glassy carbon films annealed via RTA at 800 °C, 900 °C, and 950 °C for 1 min exhibit a completely different microstructure, as shown in Figure 8.

At 800 °C, AFM analysis (Figure 8A) reveals a relatively low RMS roughness of 5.39 nm, indicating a smooth but discontinuous carbon film with visible wrinkles (highlighted by arrows). This smoothness suggests that the Ni film undergoes minimal structural changes during RTA, allowing for the formation of a bilayer graphene structure, as confirmed by Raman spectroscopy (Figure 4D–F). In contrast to the film at 700 °C, where GC dominated, GG formation is initiated at 800 °C. This observation aligns with findings by Li et al. [63], who reported that 800 °C is an optimal annealing temperature for amorphous carbon to diffuse through nickel and undergo layer exchange, facilitating graphene formation.

The I_2D_/I_G_ ratio obtained from Raman analysis aligns with values expected for bilayer graphene, confirming an improvement in structural order and quality. SEM analysis at this temperature (Figure 8B) shows characteristic wrinkle-like structures (marked by arrows), holes bordered by bright edges, and the presence of sub-5 nm Ni particles. Wrinkling is a well-documented feature of graphene growth on polycrystalline Ni, resulting from the mismatch in thermal expansion coefficients between Ni and graphene. Upon cooling, differential contraction generates compressive stress in the graphene layer, which is relieved through wrinkling [58]. Despite bilayer graphene formation, the presence of holes with bright edges indicates incomplete graphene coverage at 800 °C. These holes indicate that the thermal energy at this temperature is insufficient to fully diffuse carbon atoms across the Ni surface, leading to localized discontinuities in the graphene film. The bright edges around the holes are likely caused by edge effects in SEM imaging, where enhanced electron scattering at discontinuities increases contrast. Additionally, SEM (Figure 8C) and the corresponding energy-dispersive X-ray (EDX) line scan analysis (Figure 8D) confirm the presence of sub-10 nm Ni particles, suggesting the onset of localized dewetting. However, AFM analysis still shows a relatively smooth Ni surface, indicating that large-scale Ni agglomeration has not yet occurred.

Previous studies [41,48,52,64] reported that Ni films thinner than 50 nm tend to agglomerate during annealing, leaving residual Ni particles on the surface. Moreover, Li et al. [63] observed that 80 nm Ni films begin to agglomerate at temperatures as low as 800 °C. In this case, the small size of the Ni particles suggests that while localized dewetting has begun, the majority of the Ni layer remains structurally intact. Thus, 800 °C marks a critical transition from GC to bilayer graphene, with minimal dewetting and largely preserved Ni integrity. The observed wrinkles, holes, and Ni particles reflect the interplay between thermal energy, carbon diffusion, and Ni film stability, shaping the emerging graphene morphology.

At 900 °C, AFM analysis (Figure 8E) reveals an increase in RMS roughness to 8.46 nm, indicating significant structural changes in the Ni film, such as grain growth, partial dewetting, and increased surface irregularities. These changes are consistent with the effects of higher annealing temperatures, which enhance Ni atom mobility and promote morphological evolution [65]. SEM imaging (Figure 8F) shows that the graphene layer closely follows the underlying Ni grain structure, a hallmark of the metal-induced crystallization and layer exchange mechanism. This finding aligns with previous reports on CVD-grown graphene on polycrystalline Ni films [56]. Despite these morphological transformations, Raman analysis suggests that increasing the temperature from 800 °C to 900 °C does not significantly alter the number of graphene layers, as reflected by a slight decrease in the I_2D_/I_G_ ratio from 0.997 to 0.956. This indicates that while the graphene thickness remains relatively stable, the structural evolution of the Ni film primarily governs changes in the graphene morphology.

A notable feature in SEM images at 900 °C is the presence of bright islands within the grains, a phenomenon consistent with previous studies [29,63]. EDX mapping of Ni (inset of Figure 8F) confirms that these bright regions correspond to Ni/C phases covered by a carbon layer, further supporting the MICE mechanism, where carbon atoms diffuse into the Ni film and recrystallize into graphene layers at the surface. Interestingly, SEM imaging at different locations (Figure 8G) reveals a structural transition from a grainy morphology (spot 1) to a more continuous graphene film (spot 2). This transition is characterized by interconnected bright islands (Ni/C phase), tiny Ni particles (<5 nm), and larger residual Ni regions (~15 nm), accompanied by a darker matrix containing more carbon, as shown in higher magnification in Figure 8H. Comparing these two regions suggests that initially isolated bright islands within the grains begin to merge, forming larger, more unified islands as diffusion and recrystallization progress. Profilometer measurements between spot 1 and spot 2 reveal a step height of 173 nm, confirming the evolution of graphene morphology driven by the RTA process. Additionally, while very small Ni particles reappear, similar to those observed at 800 °C, the emergence of island-like structures and larger residual Ni particles (Figure 8H) indicate a more pronounced dewetting effect at 900 °C.

Dewetting occurs as increased thermal energy enhances Ni atom mobility, causing the film to agglomerate into isolated particles [66,67]. Miyoshi et al. [67] reported that Ni agglomeration progresses with increasing annealing temperature and is nearly complete above 900 °C due to the tendency to minimize surface energy. Consequently, this dewetting effect likely contributes to the observed decrease in the I_2D_/I_G_ ratio. This observation aligns with findings by Byan et al. [48], who reported that Ni agglomeration leads to a strong D band and a significantly weakened 2D band in Raman spectra. Moreover, wrinkle formation becomes more pronounced at 900 °C, with larger wrinkles emerging along grain boundaries and smaller wrinkles persisting inside the grains (marked by arrows). This behavior aligns with findings by Chae et al. [56], who reported that as graphene flakes grow and merge, cooling-induced strain leads to wrinkle formation as a stress-relief mechanism. The observed wrinkle patterns reflect enhanced graphene growth kinetics and the transition toward a more interconnected graphene structure at this temperature. Thus, at 900 °C, the RTA process drives substantial changes in both the Ni substrate and the graphene film. The combination of larger grains, interconnected bright islands, prominent wrinkles, and residual Ni particles reflects the complex interplay between carbon diffusion, Ni recrystallization, and dewetting, ultimately shaping the graphene morphology.

At 950 °C, AFM analysis (Figure 8I) reveals an increase in RMS roughness to 9.6 nm, reflecting the continued evolution of the graphene film and the agglomeration of the Ni layer. This roughness can be attributed to the formation of multilayer graphene, as indicated by Raman analysis, and the presence of larger residual Ni particles (≈42 nm) [64], resulting from ongoing agglomeration, as confirmed by EDX analyses (Figure 8K). Compared to 900 °C (Figure 8H), SEM images at 950 °C (Figure 8J) show larger, less bright Ni/C islands interspersed with darker regions, indicative of a thicker and more heterogeneous graphene layer. Raman spectra further support this observation, showing a reduction in the I_2D_/I_G_ ratio (Table 2), which signifies a transition from bilayer graphene at lower temperatures to multilayer graphene. This transition is consistent with the fact that higher annealing temperatures enhance carbon diffusion in Ni, facilitating the formation of thicker graphene layers upon cooling [68,69]. However, the increased temperature also accelerates structural changes in the Ni layer, particularly dewetting, which disrupts the catalytic uniformity required for consistent bilayer graphene formation. SEM images (Figure 8J) reveal larger bright Ni particles compared to those observed at 900 °C, embedded within the graphene layer. EDX mapping confirms these particles as residual Ni (Figure 8K).

The presence of larger Ni/C islands and partially embedded Ni particles at 950 °C reflects the dual effect of increased annealing temperature: while promoting the growth of thicker graphene layers, it also induces structural instabilities in the Ni layer. The progression from 800 °C to 950 °C demonstrates a clear trend: higher temperatures enhance graphene growth but simultaneously exacerbate dewetting and structural instabilities in the Ni layer. At 950 °C, these effects culminate in the formation of multilayer graphene with increased roughness, larger Ni/C islands, and residual Ni particles, highlighting the delicate interplay between temperature, Ni evolution, and graphene quality.

#### 3.2.3. Glassy Graphene on PCD

Figure 9A presents an SEM image of a polished PCD film before coating with AZ 5214E photoresist, revealing flattened faceted grains and distinct grain boundaries. The surface appears relatively smooth, as confirmed by AFM analysis (inset of Figure 9A), which measures an RMS roughness of 3.47 ± 0.48 nm. Twin boundaries, marked by arrows in Figure 9A and its inset, are visible within the diamond grains, highlighting characteristic microstructural features of PCD grown via microwave plasma chemical vapor deposition. After GC was fabricated from AZ 5214E on PCD and subsequently coated with a thin nickel film, AFM analysis reveals an increase in surface roughness to 4.66 ± 0.35 nm compared to the bare PCD film, highlighting the impact of the nickel layer. While the underlying diamond morphology remains discernible, the image appears slightly blurred compared to the inset in Figure 9A, suggesting the presence of an ultrathin masking layer. This observation confirms the formation of an ultrasmooth and thin GC film, consistent with findings on the SiO_2_/Si substrate (Figure 6A,B).

To better understand the role of the substrate in metal-induced crystallization (MIC) and the layer exchange mechanism during graphene formation, we examine GG formation on PCD at 950 °C for 3 s and 1 min, focusing on the effects of Ni dewetting, substrate roughness, and grain boundaries. The surface topography and morphology of nickel-coated glassy carbon after annealing at 950 °C for 3 s are shown in Figure 10A,B. The formation of GG, confirmed by Raman analysis, is accompanied by significant morphological changes, including a substantial increase in surface roughness (RMS = 19.39 nm). Additionally, the underlying diamond structure is no longer visible, indicating that disordered carbon has undergone structural reorganization during annealing. SEM imaging (Figure 10B) reveals a GG film with some pits, which is consistent with graphene formation on PCD as reported in a previous study [37,70].

Additionally, bright-phase regions accumulated at the grain boundaries of the diamond, suggesting localized Ni redistribution during annealing. EDX mapping of the Ni element further confirms that Ni particles are embedded beneath the GG layer, while the bright-phase regions at the grain boundaries are enriched with Ni (Figure 10C). These findings strongly support the MIC and layer exchange mechanism, in which carbon atoms initially trapped beneath the Ni film (originating from the GC layer) diffuse through the metal and crystallize into graphene on the surface, effectively replacing the metal.

Concurrently, Ni atoms migrate and preferentially accumulate at the grain boundaries, further shaping the final surface morphology. Due to the short annealing time and the rapid diffusion of Ni during the temperature ramp, only a small amount of Ni remains on the surface. This behavior aligns with the findings of Berman et al. [32], who reported that during RTA, Ni atoms diffuse rapidly through ultrananocrystalline diamond (UNCD) and settle at the bottom of the UNCD/Si interface. The accumulation of Ni at diamond grain boundaries acts as nucleation sites for multilayer graphene [32,37], increasing both the layer thickness and inhomogeneity in the number of graphene layers, leading to greater structural disorder. This is reflected in the Raman spectrum by a decreased I_2D_/I_G_ ratio and increased D-band intensity compared to GG on SiO_2_/Si.

With extended annealing to 1 min, Ni penetration into the grain boundaries continues, resulting in the disappearance of bright-phase Ni accumulations at the grain boundaries, as shown in Figure 10E. The GG layer becomes discontinuous, as evidenced by an increase in surface roughness to 28.10 nm (Figure 10D). Figure 10F provides a clearer view of small, bright Ni particles embedded within the GG islands, indicating deeper Ni diffusion beneath the graphene layer. This behavior contrasts with observations on SiO_2_/Si, where Ni dewetting leads to the formation of residual Ni particles on the surface (Figure 8J). In contrast, on PCD, Ni preferentially migrates into the grain boundaries rather than forming discrete surface islands, preventing significant dewetting. This suggests that Ni-PCD interactions stabilize the Ni layer, influencing graphene nucleation and growth mechanisms. Additionally, the microstructure of GG differs significantly between the two substrates. On SiO_2_/Si, GG appears smooth and continuous, whereas on PCD, it forms interconnected graphene islands with embedded Ni-rich particles and an agglomerated carbon network, leading to increased roughness. This structural difference suggests that grain boundaries play a crucial role in graphene evolution, resulting in a distinct morphology compared to non-polycrystalline substrates.

Wrinkled structures are observable on interconnected graphene islands (Figure 8G and Figure 10D), contributing to the D-band intensity in the Raman spectrum. This is a characteristic feature of Ni-catalyzed graphene growth. This suggests that wrinkle formation is independent of the substrate type and instead arises from the large thermal expansion mismatch between Ni and graphene during rapid cooling.

### 3.3. Electrochemical Characterization

#### 3.3.1. Electrochemical Characterization in a Supporting Electrolyte

The electrochemical potential window is a crucial parameter in determining an electrode’s suitability for electrochemical sensing. To evaluate this, cyclic voltammetry (CV) was performed on three types of electrodes: (i) GC derived from AZ 5214E on PCD at 950 °C for 1 min, (ii) GG electrodes fabricated under the same conditions on PCD, and (iii) GG electrodes fabricated at different annealing temperatures on SiO_2_/Si. Measurements were carried out in PBS over a potential range of −1.5 V to 1.5 V at a sweep rate of 0.1 V s^−1^, as shown in Figure 11, Figure 12 and Figure 13. The electrochemical potential window was determined by identifying the anodic and cathodic limits, defined as the potentials where the current density exceeded ±200 µA/cm^2^, indicating the onset of the oxygen evolution reaction (OER) and hydrogen evolution reaction (HER). A summary of the potential windows for each electrode type is provided in Table 3.

As shown in Figure 11, the CV response of GC/PCD reveals no distinct Faradaic redox processes apart from the expected HER in the cathodic region and OER in the anodic region. This behavior is consistent with previous studies, which reported that in neutral media, such as K_2_SO_4_, Faradaic peaks may not appear, whereas in acidic media, such as H_2_SO_4_, GC exhibits a Faradaic oxidation peak at approximately 0.69 V vs. Ag/AgCl due to the oxidation of surface functional groups [15,71]. This observation confirms that under the present experimental conditions, redox-active surface functionalities are not significantly electrochemically active. Furthermore, CV measurements over multiple cycles (inset of Figure 11) show no evidence of electrode degradation, indicating the stability of the GC electrode. However, GC exhibits a significantly high capacitive current, suggesting substantial double-layer charging. This enhanced capacitance is likely due to the presence of nanopores (<2 nm) or mesopores (2–50 nm), which increase the electrochemically active surface area [72] despite the film’s smooth and dense morphology, as observed in SEM and AFM analyses. Raman spectroscopy further confirms the structural disorder in GC, as evidenced by the presence of a prominent D band (Figure 3B), which indicates edge-plane defects that enhance charge storage. Additionally, residual surface functional groups such as hydroxyl, carbonyl, and carboxyl, commonly retained in PPF-derived electrodes [8], can increase the density of electronic states, further contributing to the high background current.

Upon the conversion of GC to GG on PCD, CV curves (Figure 11) reveal a significant reduction in double-layer capacitance, decreasing from 324.37 to 72.98 µF/cm^2^. The double-layer capacitance values of GG on SiO2/Si and PCD at different annealing temperatures are summarized in Table 3. This reduction is attributed to structural changes in the electrode. Raman analysis reveals a lower I_D_/I_G_ ratio (0.23 vs. 0.70 for GC/PCD), indicating the formation of larger, more ordered sp^2^ domains. This reduction in defects, particularly at basal and edge planes, decreases the content of oxygen functional groups, which are known to contribute to higher capacitance [73,74]. Despite the increased roughness of GG on PCD, the reduction in oxygen functionality likely plays a dominant role in lowering the double-layer capacitance. While this transition reduces the electrode’s charge storage capacity, it enhances the electrochemical sensing performance by lowering background capacitive currents and improving the signal-to-noise ratio.

Interestingly, the absence of significant redox activity between OER and HER suggests that the embedded Ni-rich particles within the GG matrix are either electrochemically inactive or encapsulated beneath graphene layers. This interpretation is consistent with SEM observations (Figure 10E,F), indicating that these particles have limited direct involvement in charge transfer. However, a minor shoulder around −0.7 V, likely corresponding to the oxygen reduction reaction (ORR), suggests that embedded Ni-rich particles enhance sluggish ORR kinetics, even in deaerated PBS. Compared to GC, GG exhibits significantly higher anodic currents for OER, which can be attributed to the presence of embedded Ni-rich particles. Nickel is a well-known catalyst for OER [75], and its incorporation within the rough graphene matrix enhances charge transfer kinetics while increasing the density of catalytically active sites. The increased roughness, as revealed by AFM, further amplifies the electrochemically active surface area (ECSA), thereby facilitating enhanced OER activity. Despite the differences in OER behavior, the cathodic current corresponding to HER remains similar for both films. This suggests that the catalytic contribution of Ni in GG is more pronounced for OER than for HER, likely due to differences in reaction kinetics and surface accessibility. While Ni is known to catalyze HER [76], its embedded nature within the graphene matrix may restrict its electrochemical activity, leading to no significant improvement in HER performance compared to GC. However, the increased ECSA compensates for the lower catalytic contribution of embedded Ni, leading to comparable cathodic current densities. Overall, the electrochemical characterization, supported by microstructural and spectroscopic analyses, demonstrates that the transformation from GC to GG results in reduced double-layer capacitance and enhanced catalytic properties for OER due to the incorporation of Ni-rich particles. The increased surface roughness and interconnected graphene island morphology in GG on PCD further contribute to its superior OER performance, highlighting its potential as a functional electrode material for electrocatalysis.

##### Effect of Annealing Temperature on the Electrochemical Properties of Glassy Graphene Microelectrodes on SiO_2_/Si

The effect of annealing temperature on the electrochemical properties of GG on SiO_2_/Si was evaluated by measuring its accessible potential window, as shown in Figure 12A.

As the annealing temperature increases from 800 °C to 950 °C, the potential window narrows (as reported in Table 3), indicating enhanced electrochemical activity. This can be attributed to microstructural and structural modifications, as confirmed by Raman spectroscopy and SEM. At higher temperatures, the transition from bilayer to multilayer graphene occurs, increasing the density of active sites for heterogeneous electron transfer (HET) and contributing to the narrowing of the potential window. Graphene edge planes are known to exhibit faster HET compared to basal planes [22], further explaining this trend. At 950 °C, SEM reveals large residual Ni particles and Ni/C islands, which enhance charge transfer kinetics and facilitate HER and OER at lower overpotentials. Despite this, no Ni-related redox peaks are observed, suggesting that the residual Ni is either electrochemically inactive, potentially due to oxidation, or partially encapsulated within the graphene matrix, limiting its direct electrochemical contribution. However, a minor cathodic shoulder before HER, more pronounced at 950 °C, likely corresponds to ORR, similar to what was observed on PCD (Figure 11). Due to instrumental limitations, a misleading plateau appears for OER at higher potentials. Interestingly, the GG electrode annealed at 900 °C exhibits a higher cathodic current density than the 950 °C sample, likely due to the combination of a grainy structure and a more continuous graphene film, as observed in SEM (Figure 8G), which may optimize charge transport pathways. Additionally, the C_dl_ increases slightly from 800 °C to 900 °C but nearly doubles at 950 °C, reaching 68.67 µF/cm^2^ (Table 3). It has been reported that graphene capacitance depends on the number of layers and the amount of active area, particularly sp^2^ edge planes and defects [18]. Thus, our results suggest that both multilayer graphene formation and increased roughness contribute to the enhanced C_dl_, as confirmed by AFM analysis.

##### Influence of Substrate on the Electrochemical Performance of Glassy Graphene Microelectrodes

The electrochemical behavior of GG fabricated at 950 °C was compared on two different substrates: PCD and SiO_2_/Si. CV measurements reveal that GG on PCD exhibited a wider potential window compared to SiO_2_/Si (Figure 12B). This difference can be attributed to variations in the microstructure and Ni distribution, as confirmed by SEM and Raman spectroscopy. On SiO_2_/Si, larger residual Ni and Ni/C islands could introduce localized catalytic effects, leading to additional side reactions that reduce the potential window. In contrast, on PCD, Ni particles are smaller and more embedded, reducing their direct electrochemical impact and thus maintaining a slightly wider electrochemical window. Additionally, the higher HER and OER currents observed for GG on SiO_2_/Si compared to PCD suggest that residual Ni on this substrate plays a more active catalytic role in hydrogen and oxygen evolution. This enhanced activity likely stems from the greater exposure of Ni on the surface, as opposed to the Ni-rich particles embedded within the rough graphene matrix on PCD. The increased accessibility of Ni sites on SiO_2_/Si facilitates more efficient charge transfer, thereby promoting electrocatalytic reactions. Despite the wider potential window, GG on PCD exhibits a slightly higher double-layer capacitance (72.98 vs. 68.67 µF/cm^2^) than on SiO_2_/Si. This suggests that increased surface roughness, more embedded Ni-rich particles, and the formation of thicker multilayer graphene contributed significantly to C_dl_. These findings highlight the dual role of residual Ni: exposed Ni on SiO_2_/Si enhances HER, OER, and ORR but narrows the potential window, while embedded Ni-rich particles on PCD contribute less to HER and OER but increase C_dl_, particularly in combination with multilayer graphene formation.

In addition, unlike GG on PCD, GG on SiO_2_/Si is more prone to delamination during electrochemical cycling. This instability is likely due to weaker adhesion between the graphene layer and the SiO_2_ substrate, which lacks strong interfacial bonding. In contrast, PCD provides a more compatible carbon–carbon interface, enhancing mechanical stability and electrode durability under repeated electrochemical cycling.

#### 3.3.2. Electrochemical Characterization with Redox Marker

##### Outer-Sphere Redox Marker

The electrochemical performance of the fabricated GC and GG electrodes was further evaluated using CV with two redox probes, [Ru(NH_3_)_6_]^3+/2+^ and [Fe(CN)_6_]^3−/4−^, in 0.1 M KNO_3_. Table 4 summarizes the anodic and cathodic peak potential difference (ΔE_p_), which serves as an indicator of electrode quality by reflecting electron transfer kinetics and redox reversibility, along with the anodic-to-cathodic peak current density ratio (I_p_^a^/I_p_^c^).

[Ru(NH3)6]^3+/2+^ serves as a representative outer-sphere redox marker, meaning its electron transfer (ET) process is governed primarily by simple diffusion and is minimally affected by surface chemistry, such as functional groups or surface termination [77]. This makes it particularly useful for evaluating the intrinsic electrochemical response of electrodes exposed to air.

As shown in Figure 13A, GG exhibits more quasi-reversible redox behavior than GC, with higher peak currents, a narrower ΔE_p_ (78 mV vs. 163 mV), and a more symmetric voltammetric profile. These improvements stem from a higher density of sp^2^-hybridized carbon domains, as confirmed by Raman spectroscopy, and enhanced conductivity, which facilitate efficient charge transfer. Additionally, GG electrodes demonstrate excellent durability over multiple electrochemical cycles, suggesting good structural stability (inset of Figure 13A).

Figure 13B presents the voltammetric responses of [Ru(NH_3_)_6_]^3+/2+^ for GG electrodes fabricated at different annealing temperatures on the SiO_2_/Si substrate. As the annealing temperature increases from 800 °C to 900 °C, peak currents rise and ΔE_p_ narrows (Table 4), indicating improved ET kinetics, while I_p_^a^/I_p_^c^ remains relatively constant. This improvement is attributed to the growth of more ordered sp^2^ carbon domains, as confirmed by Raman spectroscopy (Table 2), along with microstructural changes that lead to the formation of GG with fewer defects. These structural refinements influence the electronic properties of the carbonaceous film, enhancing charge transport and reducing electrical resistance. However, at 950 °C, despite the formation of additional multilayer graphene, increased surface roughness, and the presence of larger residual Ni and Ni/C islands, ΔE_p_ unexpectedly widens. This suggests that interfacial resistance (iR) may become a limiting factor at higher annealing temperatures, where weaker adhesion and increased resistive effects hinder charge transport. Unlike studies that apply iR compensation to eliminate interfacial resistance effects, our CV measurements retain these contributions to provide a more comprehensive evaluation of charge transfer kinetics and electrode–substrate interactions. The impact of interfacial resistance will be further examined in EIS measurements.

A comparison of CV measurements for GG fabricated at 950 °C for 1 min on SiO_2_/Si and PCD (inset of Figure 13B) reveals enhanced HET kinetics on PCD, evidenced by a narrower ΔE_p_ This improvement is attributed to the larger electrochemically active surface area resulting from the higher surface roughness, as well as the formation of a thicker, more structurally integrated graphene layer. Additionally, the higher I_p_^a^/I_p_^c^ ratio on PCD (0.70 vs. 0.57 on SiO_2_/Si) suggests a more efficient and reversible electron transfer, likely due to a stronger adhesion between the graphene layer and the PCD substrate. This stronger interaction reduces interfacial resistance and improves long-term electrochemical stability. In contrast, although the thinner graphene layer on SiO_2_/Si contains fewer defects, its weaker interaction with the substrate leads to a progressive delamination over multiple CV cycles, resulting in an increased ΔE_p_ and a decline in electrochemical performance. These findings suggest that, beyond the delamination issues observed on SiO_2_/Si, the stronger adhesion and structural integrity of graphene on PCD play a crucial role in facilitating charge transfer and ensuring long-term stability.

##### Inner-Sphere Redox Marker

The [Fe(CN)_6_]^3−/4−^ redox pair is a well-established inner-sphere redox system on carbon-based electrodes, where ET is significantly influenced by surface chemistry, including defects and functional groups, which can serve as active sites [2]. Unlike outer-sphere redox markers, which primarily depend on diffusion-controlled HET, the response of [Fe(CN)_6_]^3−/4−^ is highly sensitive to surface properties, making it a valuable probe for assessing electrode surface modifications. As shown in Figure 13C, GG exhibits well-defined redox peaks with a narrower ΔE_p_ compared to GC, indicating enhanced HET kinetics, similar to the trends observed for [Ru(NH_3_)_6_]^3+/2+^ (Table 4). The decrease in ΔE_p_ for GG compared to GC suggests that, in addition to the formation of extended sp^2^ domains, the lower oxygen functionalization plays a key role in facilitating the charge transfer for the inner-sphere [Fe(CN)_6_]^3-/4-^ redox couple. This is further supported by the lower **C_dl_** of GG, which reflects a cleaner surface with fewer functional groups that could hinder ET. However, despite the enhanced charge transfer kinetics on GG, the ΔE_p_ for [Fe(CN)_6_]^3−/4−^ remains higher than that for the outer-sphere [Ru(NH₃)₆]^3+^/^2+^ redox system. This confirms that oxygen functional groups hinder ET by blocking adsorption sites and increasing charge transfer resistance. Additionally, [Fe(CN)_6_]^3−/4−^ exhibits a higher I_p_^a^/I_p_^c^ ratio, suggesting an increased surface sensitivity. This effect may arise from charge repulsion interactions, surface inhomogeneities, or partial adsorption effects, which are common for inner-sphere redox processes. Furthermore, for the [Fe(CN)_6_]^3−/4−^ redox couple, the CV curves remained consistent over five cycles with negligible variation in ΔE_p_, similar to the outer-sphere [Ru(NH_3_)_6_]^3+/2+^ system. This stability suggests that the electrode surface remains chemically and structurally stable under repeated redox cycling, with minimal adsorption or surface reconstruction effects.

The voltammetric responses of [Fe(CN)_6_]^3−/4−^ on GG electrodes fabricated at different annealing temperatures on SiO_2_/Si show minimal variation in ΔE_p_ across 800 °C, 900 °C, and 950 °C (Figure 13D), in contrast to the reduction observed for the outer-sphere [Ru(NH_3_)_6_]^3+/2+^ system. This suggests that for inner-sphere reactions, surface chemistry, particularly oxygen functional groups, plays a more dominant role in controlling HET than microstructural changes. While annealing enhances graphitization and sp^2^ carbon domains, the persistent influence of oxygen functionalities limits HET improvement, reinforcing their role as primary charge transfer barriers in inner-sphere systems.

Interestingly, while GG on PCD exhibits enhanced HET kinetics for the outer-sphere [Ru(NH_3_)_6_]^3+/2+^ system compared to GG/SiO_2_/Si, the HET rate for the inner-sphere [Fe(CN)_6_]^3−/4−^ remains comparable on both substrates (inset of Figure 13D). This suggests that while microstructural changes and higher conductivity strongly influence outer-sphere redox reactions, their effect on inner-sphere processes is less pronounced. Despite this, the rougher morphology of GG on PCD increases the electrochemically active surface area, leading to higher peak currents despite similar ΔE_p_ values. This suggests that, for inner-sphere redox couples, surface chemistry, particularly the presence of oxygen functional groups, plays a more dominant role in governing HET kinetics than substrate-induced microstructural and conductivity changes.

#### 3.3.3. Electrochemical Impedance Spectroscopy (EIS)

To further investigate the charge transfer and electrochemical behavior of GG on different substrates, EIS measurements were performed in 1 mM [Ru(NH_3_)_6_]^3+/2+^ with 0.1 M KNO_3_. The impedance data were fitted using the equivalent circuit [R_s_ (CPE_1_(R_1_W))(CPE_2_R_2_)], which met the fitting criteria of χ^2^ < 0.01 for both PCD and SiO_2_/Si. In this model, R_s_ represents the solution resistance, while CPE_1_ is a constant phase element that accounts for the capacitance at the GG–electrolyte interface. The resistance R_1_ corresponds to the interfacial resistance, which reflects the charge transfer characteristics at the graphene–substrate interface and is influenced by adhesion and substrate interaction. The Warburg impedance (W) describes mass transport effects in the electrochemical system. Additionally, CPE_2_ represents a second constant phase element, which captures the capacitive contribution from electroactive sites, and R_2_ corresponds to the charge transfer resistance associated with these electroactive sites.

The EIS results reveal distinct differences in charge transfer behavior between GG fabricated at 950 °C for 1 min on SiO_2_/Si and PCD, highlighting the impact of substrate properties on electrochemical performance. The Nyquist plot of GG/PCD does not exhibit an observable semicircle at high frequencies, suggesting a highly efficient charge transfer process with minimal interfacial resistance (Figure 14A). In contrast, GG/SiO_2_/Si (Figure 14B) displays a prominent semicircle at high frequencies and an even larger one at lower frequencies, indicative of significant interfacial and charge transfer resistance.

The Bode phase plot for SiO_2_/Si exhibits two distinct peaks, suggesting the presence of two electrochemically active sites (inset of Figure 14B). This observation aligns with SEM analysis, which reveals large residual Ni and Ni/C clusters along with a graphene phase. In contrast, the Bode phase plot for PCD shows a single dominant process, indicating more uniform charge transfer characteristics (inset of Figure 14A).

A quantitative analysis of the resistance values supports these observations. On SiO_2_/Si, the extracted resistance values show that R_1_ (1479 Ω cm^2^) is significantly higher than that on PCD (878 Ω cm^2^), indicating that weaker adhesion at the graphene–SiO_2_/Si interface increases interfacial resistance and hinders charge transport. This poor adhesion likely contributes to the delamination of the films over multiple CV cycles, further confirming its impact on electrochemical stability. The impact of this R_1_ is evident in the CV data, where the broader ΔE_p_ (90 mV) compared to PCD (78 mV) suggests slower electron transfer kinetics. Since the recorded CV curves were obtained without iR compensation, the higher R_1_ on SiO_2_/Si directly contributes to the increased overall resistance, reinforcing the conclusion that poor interfacial contact dominates the electrochemical response.

In contrast, the lower R_1_ (878 Ω cm^2^) on PCD indicates stronger adhesion and improved electrical coupling between graphene and the substrate, enhancing charge transfer efficiency.

A notable difference between the two substrates is the charge transfer resistance associated with electroactive sites (R_2_), which is higher on PCD (21.88 Ω cm^2^) than on SiO_2_/Si (11.11 Ω cm^2^). This increase can be attributed to the deeper embedding of Ni particles within the graphene matrix on PCD, as confirmed by SEM analysis. On SiO_2_/Si, the more exposed and partially embedded Ni clusters create localized charge transfer regions, as evidenced by an additional peak (θ = 43°) at 100 Hz in the Bode phase plot. This leads to a lower R_2_ further supported by the contribution of the graphene phase with fewer defects. However, despite the lower R_2_, the overall electrochemical performance remains hindered by the high R_1_ and poor adhesion, which limit charge transport and long-term stability. The impact of high R_1_ on SiO_2_/Si is evident in the Nyquist plot, where the increased interfacial resistance leads to a pronounced semicircle, indicating hindered charge transfer. In contrast, the stronger adhesion of GG on PCD leads to lower R_1_, reducing interfacial resistance and facilitating charge transport. This is reflected in the Nyquist plot by the absence of a large semicircle, despite the slightly higher R_2_. These findings highlight that charge transfer resistance is strongly influenced by R_1_, emphasizing the critical role of substrate adhesion in optimizing electrochemical performance.

#### 3.3.4. Voltammetric Detection of Adrenaline

To assess the electrochemical sensing capability of the fabricated GG electrodes, CV measurements were performed using 1 mM adrenaline in PBS at pH 7.4 and a scan rate of 100 mV s^−1^. The electrochemical response was highly reproducible across all four electrodes on each GG chip, confirming the consistency of the sensing platform. Representative CV curves for GG/PCD are shown in Figure 15.

The CV measurements of GG/PCD exhibit irreversible electrochemical behavior, characteristic of catecholamine oxidation due to their two-electron, two-proton oxidation mechanism in physiological media (pH 7.36). Adrenaline oxidation to adrenaline-o-quinone (epinephrine-o-quinone) occurs at +0.43 V in the forward scan.

However, in the reverse scan, no distinct reduction peak for adrenaline-o-quinone is observed. Instead, a significantly less intense peak at −0.43 V emerges, suggesting that secondary redox-active compounds are formed during oxidation, rather than direct reduction of adrenaline-o-quinone. This behavior aligns with previous studies on adrenaline oxidation on GC electrodes [78,79]. Despite the well-documented tendency of catecholamine oxidation products to induce electrode fouling through polymerization [16], GG/PCD demonstrates remarkable stability with minimal signal loss over multiple cycles. This suggests that GG possesses a reduced number of adsorption sites, preventing excessive accumulation of oxidation byproducts and maintaining electrode performance.

As shown in the inset of Figure 15A, the oxidation peak current for 1 mM adrenaline increases linearly (R^2^ = 0.986 ± 0.003) with the square root of the scan rate, indicating a diffusion-controlled electrode reaction. This observation reinforces the hypothesis that glassy graphene’s surface properties mitigate electrode passivation, ensuring stable electrochemical performance.

Figure 15B compares the CV responses of GG electrodes fabricated on SiO_2_/Si at different annealing temperatures under identical conditions. Electrodes annealed at 800 °C and 900 °C exhibit lower sensitivity compared to those fabricated at 950 °C, where the oxidation peak current is significantly enhanced, and the oxidation potential shifts by 0.17 V toward lower values. This catalytic effect can be attributed to the presence of large residual Ni and Ni/C islands, which act as electroactive sites facilitating electron transfer.

Notably, an additional pair of voltammetric peaks at negative potentials appears, corresponding to the adrenochrome/leucoadrenochrome redox system. This behavior is consistent with the CV response of adrenaline on freshly cleaved highly oriented pyrolytic graphite (HOPG), a widely accepted model for graphene-based surfaces [80]. The clearer redox peaks observed on GG/SiO_2_/Si suggest that adrenochrome is more electrochemically active on this substrate, possibly due to differences in Ni distribution and graphene surface properties. A further comparison between GG/PCD electrodes fabricated at 950 °C for 3 s and 1 min, along with GG/SiO_2_/Si (Figure 15C), provides additional insights into the role of Ni in the electrochemical response. The enhanced oxidation signal observed for GG/PCD fabricated at 950 °C for 3 s and GG/SiO_2_/Si can be attributed to the presence of more exposed residual Ni, which reduces the overpotential for adrenaline oxidation and improves sensitivity compared to GG/PCD fabricated at 950 °C for 1 min due to differences in microstructure. However, the improved performance of GG/PCD may not solely be due to Ni catalysis. The rougher interfacial layer in GG/PCD increases the conductive area of the electrode, leading to an overall enhancement in electrochemical activity due to its higher specific surface area. This interplay between Ni dispersion, graphene structure, and substrate interactions highlights the critical role of processing conditions in optimizing electrochemical sensor performance.

#### 3.3.5. Microfluidic Detection of Adrenaline

Due to the delamination of GG/SiO_2_/Si electrodes over multiple cycles, GG/PCD fabricated at 950 °C for 3 s and 1 min were selected as the preferred platforms for flow injection amperometric (FIA) detection. Flow injection analysis, coupled with amperometric detection, was employed to evaluate the limit of detection (LOD) and sensitivity of GG microelectrodes across a concentration range of 3 µM–1 mM. Given the well-documented tendency of sp^2^ carbon electrodes to undergo fouling during catecholamine detection compared to BDD electrodes, our experimental approach incorporated extended interaction times. This strategy allowed for a thorough investigation of electrode fouling while ensuring stable and reproducible responses for each concentration without interruptions or the need for repeated injections. Figure 16A,B present the chronoamperometric responses of GG/PCD sensors fabricated at 950 °C for 3 s and 1 min at varying adrenaline concentrations in PBS. A custom-designed LabVIEW program enabled real-time, simultaneous monitoring of the amperometric signal at four different biasing voltages across the four GG electrodes on each chip. Utilizing this feature, measurements were recorded at applied voltages of 0.8 V, 0.6 V, 0.4 V, and 0.2 V on electrodes 1, 2, 3, and 4, respectively.

As illustrated in Figure 16A,B, the amperometric signal increases with higher applied voltage, reflecting the stronger driving force for electron transfer. This enhancement arises from the reduction in activation energy for adrenaline oxidation, which facilitates a more rapid electron exchange and results in a greater current response. Notably, as soon as the injection ceases, the current signal drops sharply, emphasizing the transient nature of the response. This rapid decay highlights the effectiveness of hydrodynamic conditions in FIA compared to stationary diffusion-based methods, as it significantly improves the mass transport of electroactive species to the electrode surface. During the washing step with PBS, a brief secondary increase in current was detected at lower adrenaline concentrations across all GG chips, due to residual adrenaline momentarily re-entering the detection zone before being rapidly diluted.

Interestingly, at 1 mM, a progressive decline in current was observed over 146 s for both GG/PCD electrodes fabricated at 950 °C for 3 s and 1 min. This behavior contrasts with our previous study on adrenaline detection using BDD electrodes, where no such effect was observed, due to the absence of sp^2^ carbon [42]. The observed signal decay suggests partial electrode passivation at high concentrations, potentially caused by the accumulation of oxidation byproducts on the electrode surface. At lower adrenaline concentrations, these byproducts may be present in minimal amounts and do not significantly affect the current response. However, at 1 mM, the higher local concentration increases the likelihood of strong adsorption of oxidation intermediates, progressively blocking active sites and reducing the current. Notably, the progressive decline in current is more pronounced at 0.6 V and 0.8 V. This can be attributed to the higher oxidation rate of adrenaline at elevated potentials, leading to an increased formation of reaction byproducts. At these voltages, the stronger driving force enhances electron transfer but also accelerates the generation of oxidation intermediates, which may adsorb onto the electrode surface and hinder further electron transfer. This observation differs from the CV results, where the peak current remained stable over multiple cycles at 1 mM adrenaline, as shown in the inset of Figure 15B. This suggests that in CV, the continuous potential sweeping facilitates the desorption of oxidation byproducts, preventing excessive accumulation and maintaining a steady response. In contrast, while FIA ensures a constant influx of fresh analyte, it does not actively remove adsorbed species from the electrode surface. At high concentrations (1 mM), this limitation can lead to the accumulation of reaction intermediates or byproducts on active sites, contributing to the transient passivation observed in the amperometric measurements.

Importantly, the GG chip fabricated at 950 °C for 1 min exhibits clear, steady amperometric signals over 146 s at 3 µM across all applied voltages (inset of Figure 16B), underscoring its high sensitivity and stability. At lower applied potentials (0.2 V and 0.4 V), the current remains steady throughout the entire measurement period, suggesting minimal interference from surface reactions or background drift. In contrast, at higher voltages (0.6 V and 0.8 V), the amperometric response initially gradually decreases and reaches a steady state after approximately 1 min. This delayed stabilization is likely due to transient adsorption effects or surface reorganization before diffusion-limited steady-state conditions dominate. Additionally, the exceptionally low background current density noise (300 nA/cm^2^) enables the detection of 3 µM adrenaline even at low voltages, reinforcing the superior quality of GG electrodes for sensitive and reliable electrochemical detection.

Figure 16C,D present the individual calibration curves for both GG sensors, derived from the corresponding amperograms. The current density values represent the mean steady-state current recorded during perfusion. Due to the observed fouling effects at 1 mM, the calibration curves are limited to the concentration range of 3 µM to 300 µM to ensure accurate and reliable quantification. The graphs reveal a clear increase in amperometric signal with rising adrenaline concentrations, reflecting the higher availability of electroactive molecules at the electrode surface, which enhances the oxidation reaction. Notably, both GG sensors demonstrate excellent linearity (R^2^ = 0.99) across all applied voltages within this range. This remarkable linear response can be attributed to the uniform electrode surface, efficient electron transfer kinetics, and a stable electrochemical environment, which collectively contribute to the high sensitivity and reproducibility of the GG sensors. The strong correlation between current response and concentration further underscores the reliability of these electrodes for quantitative adrenaline detection.

Figure 17A,B present the sensitivity of both GG sensors, determined by the slope of their calibration curves, and the LOD, calculated using LOD = (3 × σ)/S, where σ represents the standard deviation of the baseline noise and S is the slope of the calibration curve. Additionally, Figure 17C illustrates the signal-to-noise ratio (S/R) at 3 µM adrenaline, providing a comprehensive assessment of the sensor’s overall performance.

With increasing applied voltage, the GG sensors exhibit enhanced sensitivity and lower LOD due to the accelerated electron transfer kinetics, which facilitate more efficient oxidation of adrenaline. However, the S/N analysis reveals distinct trends between the two GG electrodes. The GG sensor fabricated at 950 °C for 3 s demonstrates a higher S/N ratio from 0.2 V to 0.6 V, whereas at 0.8 V, the GG sensor fabricated at 950 °C for 1 min exhibits superior S/N performance. The higher S/N ratio observed in the 3 s GG sensor at lower voltages (0.2 V to 0.6 V) can be attributed to its microstructural features, particularly the presence of more exposed Ni at the surface, which enhances catalytic activity and promotes efficient electron transfer. However, at 0.8 V, the combination of higher applied voltage and the catalytic role of Ni accelerates adrenaline oxidation, leading to increased accumulation of its oxidation byproducts. This, in turn, results in partial passivation of electroactive sites, causing a decline in the S/N ratio. Conversely, the 1 min GG sensor, with Ni particles more deeply embedded within the graphene matrix, mitigates excessive surface accumulation of oxidation byproducts. This results in enhanced sensitivity, a lower LOD, and a higher S/N ratio at 0.8 V.

In a comparison of these performance metrics, the GG sensor fabricated at 950 °C for 1 min demonstrates significantly higher sensitivity, a lower LOD across the voltage range of 0.2–0.8 V, and a superior S/N ratio at 0.8 V. The highest sensitivity and lowest LOD are achieved at 0.8 V, with values of 1.029 µA cm^−2^/µM and 1.05 µM, respectively, and an S/N ratio of 8.73. Table 5 compares the analytical performance of GG/PCD for adrenaline detection with other electrode materials. Compared to BDD sensors under the same detection conditions [42], GG exhibits enhanced sensitivity. However, its LOD is higher due to the increased background current density, which arises from the structural characteristics of GG compared to BDD. Interestingly, despite DPV being widely recognized for its high sensitivity [81], our GG/PCD sensor integrated with FIA outperforms many modified GC and graphene-based electrodes in terms of sensitivity. This enhanced performance can be attributed to the improved mass transport in FIA and the unique electrochemical properties of the GG/PCD electrode, which collectively facilitate electron transfer and provide a better overall detection efficiency.

Our findings highlight that the electrochemical performance of GG-based sensors for adrenaline detection is significantly influenced by Ni distribution and its interaction with the graphene matrix under FIA.

## 4. Conclusions

In this study, we developed a cost-effective, scalable, and sustainable process for fabricating GG-based sensors from a common photoresist on SiO_2_/Si and PCD substrates using rapid thermal annealing (RTA) with a thin Ni film catalyst at 800–950 °C under a nitrogen atmosphere. Raman analysis confirmed the formation of GG, while the annealing temperature and substrate type significantly influenced surface roughness, microstructure, and graphene layer thickness. On SiO_2_/Si, increasing the temperature to 950 °C enhanced Ni dewetting, leading to multilayer graphene (≥3 layers) with larger Ni/C islands and a relatively smooth surface (RMS ≈ 9 nm). In contrast, on PCD, Ni preferentially migrated into grain boundaries, resulting in smaller, embedded Ni particles and a rougher surface (RMS ≈ 28 nm) with greater inhomogeneity in graphene layer thickness. Electrochemical characterization revealed that increasing the annealing temperature from 800 °C to 950 °C reduced the potential window of GG on SiO_2_/Si from 2.10 V to 1.49 V, while increasing the double-layer capacitance from 32.48 to 68.67 µF cm^−2^. Interestingly, compared to SiO_2_/Si, GG fabricated at 950 °C on PCD exhibited a wider potential window (1.68 V vs. 1.49 V), a higher capacitance (72.98 µF cm^−2^ vs. 68.67 µF cm^−2^), and improved electron transfer kinetics, as evidenced by a lower ΔE_p_ (≈78 mV) for the [Ru(NH_3_)_6_]^3+/2+^ redox marker. These enhancements are attributed to better integration of graphene with the substrate, leading to reduced interfacial resistance, as confirmed by EIS. However, both substrates exhibited similar electron transfer behavior for the surface-sensitive [Fe(CN)_6_]^3−/4−^ redox couple, suggesting that oxygen functional groups play a dominant role in electron transfer kinetics.

For adrenaline detection, GG/SiO_2_/Si exhibited a lower oxidation overpotential (≈0.17 V) compared to GG/PCD, likely due to charge transfer differences arising from microstructural variations. However, its weaker glassy graphene–SiO_2_/Si adhesion, reflected by a higher interfacial resistance, led to delamination. Consequently, GG/PCD demonstrated superior stability and reliability as a sensing platform. FIA on GG/PCD sensors further revealed that Ni distribution and its interaction with the graphene matrix significantly impacted the sensor performance. The GG-based sensors achieved high sensitivity (1.02 µA cm^−2^/µM) over a linear concentration range of 3–300 µM and a low detection limit of 1.05 µM, highlighting their potential for high-performance electrochemical sensing. These findings suggest that PCD substrates offer a more robust platform for electrochemical sensing compared to SiO_2_/Si due to their stronger graphene–substrate integration.

## Figures and Tables

**Figure 1 sensors-25-02454-f001:**
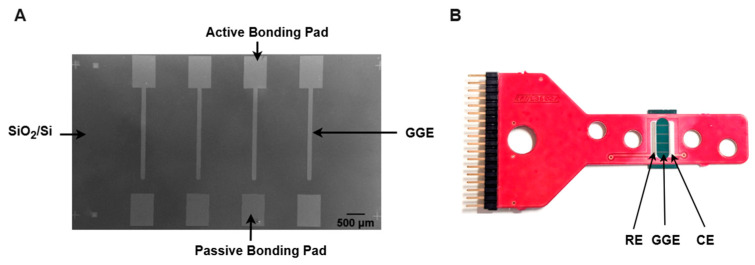
(**A**) SEM image of a diced glassy graphene (GG) chip with four stripe electrodes. (**B**) Assembled GG chip, with GGE, RE, and CE representing the glassy graphene electrode, reference electrode, and counter electrode, respectively.

**Figure 2 sensors-25-02454-f002:**
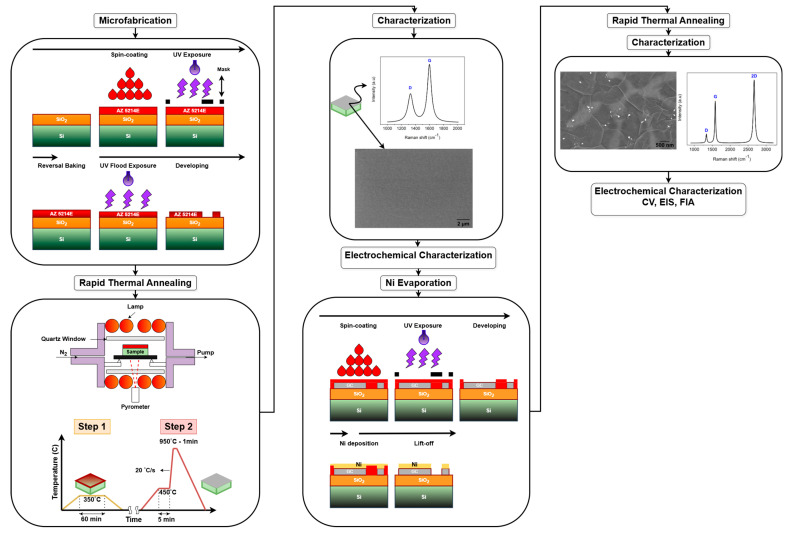
Schematic overview of the fabrication and characterization process.

**Figure 3 sensors-25-02454-f003:**
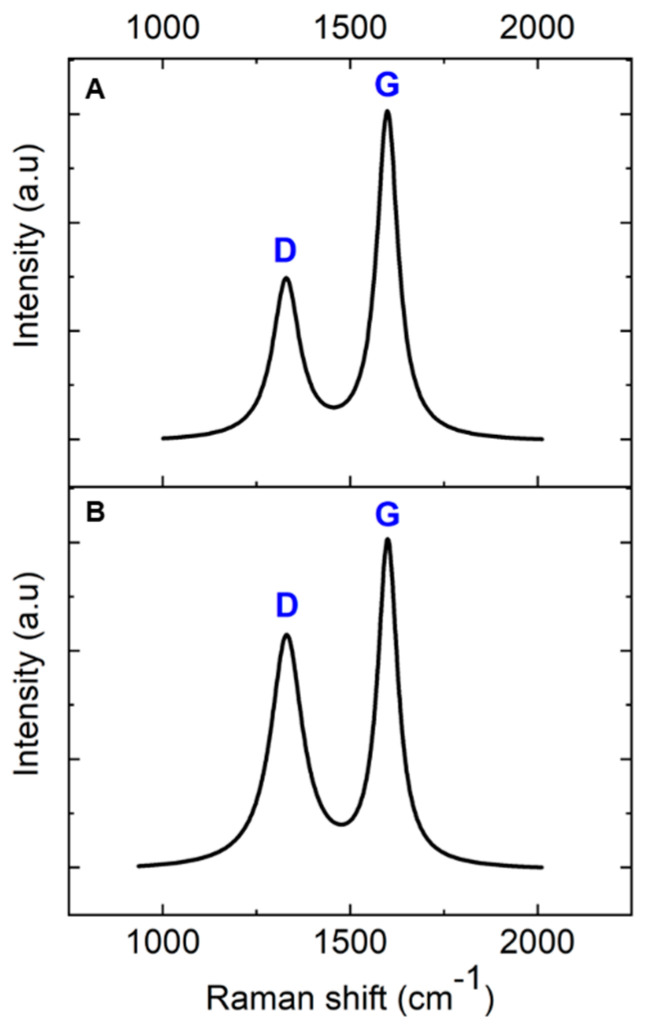
Baseline-corrected and normalized Raman spectra of pyrolyzed AZ 5214E photoresist film at 950 °C for 1 min on (**A**) SiO_2_/Si substrate and (**B**) PCD substrate.

**Figure 4 sensors-25-02454-f004:**
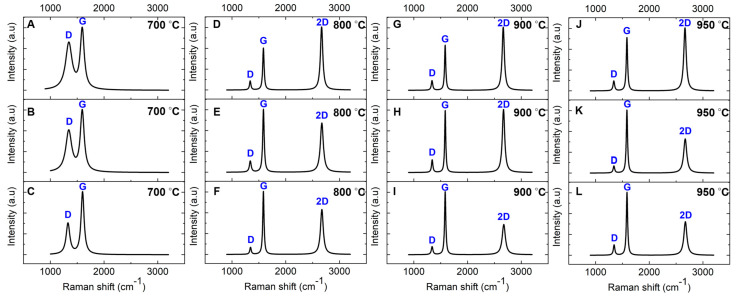
Baseline-corrected and normalized Raman spectra of nickel-coated glassy carbon films annealed via RTA for 1 min on SiO_2_/Si substrate. The spectra were measured at three distinct locations on a single sample for each temperature: (**A**–**C**) 700 °C, (**D**–**F**) 800 °C, (**G**–**I**) 900 °C, and (**J**–**L**) 950 °C.

**Figure 5 sensors-25-02454-f005:**
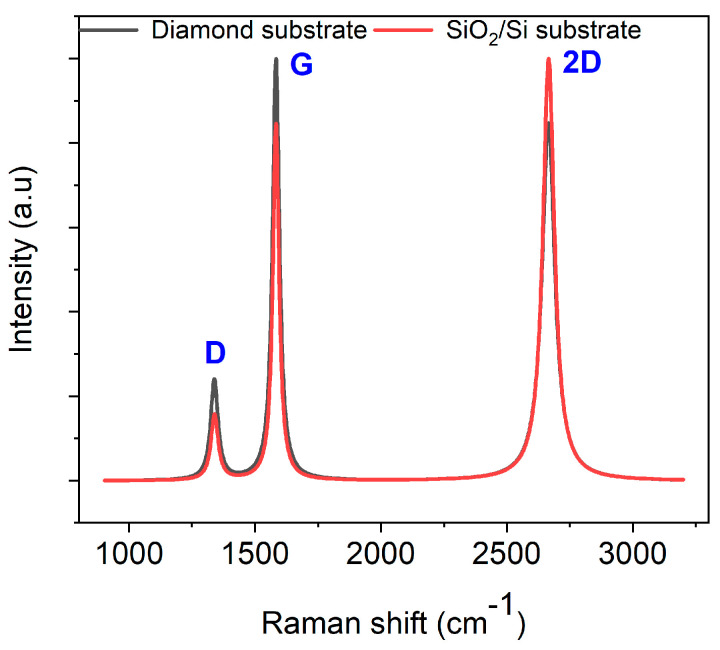
Baseline-corrected and normalized Raman spectra of nickel-coated glassy carbon films annealed at 950 °C via RTA for 1 min on polycrystalline diamond and SiO_2_/Si substrates.

**Figure 6 sensors-25-02454-f006:**
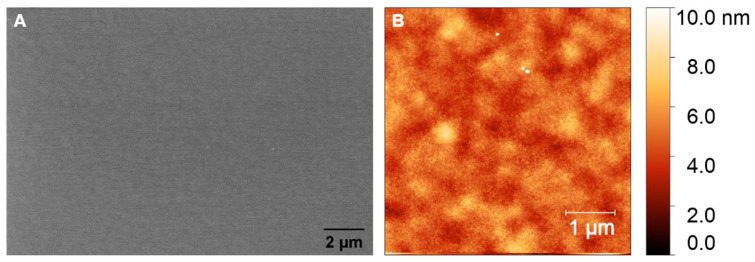
(**A**) SEM and (**B**) AFM images of GC fabricated from AZ 5214E via RTA at 950 °C for 1 min on SiO_2_/Si substrate.

**Figure 7 sensors-25-02454-f007:**
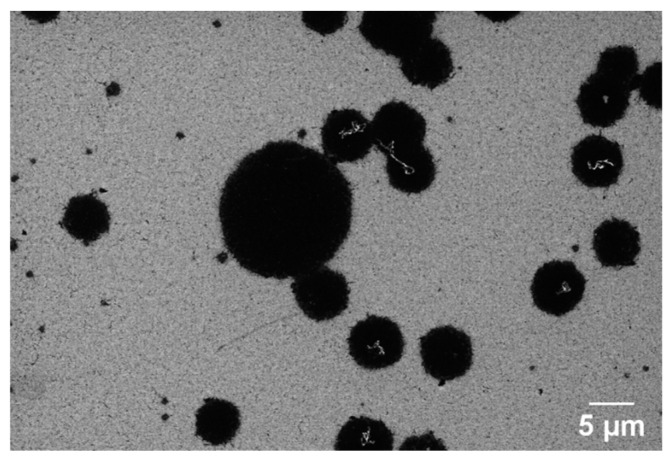
The SEM image of nickel-coated glassy carbon films annealed via RTA at 700 °C.

**Figure 8 sensors-25-02454-f008:**
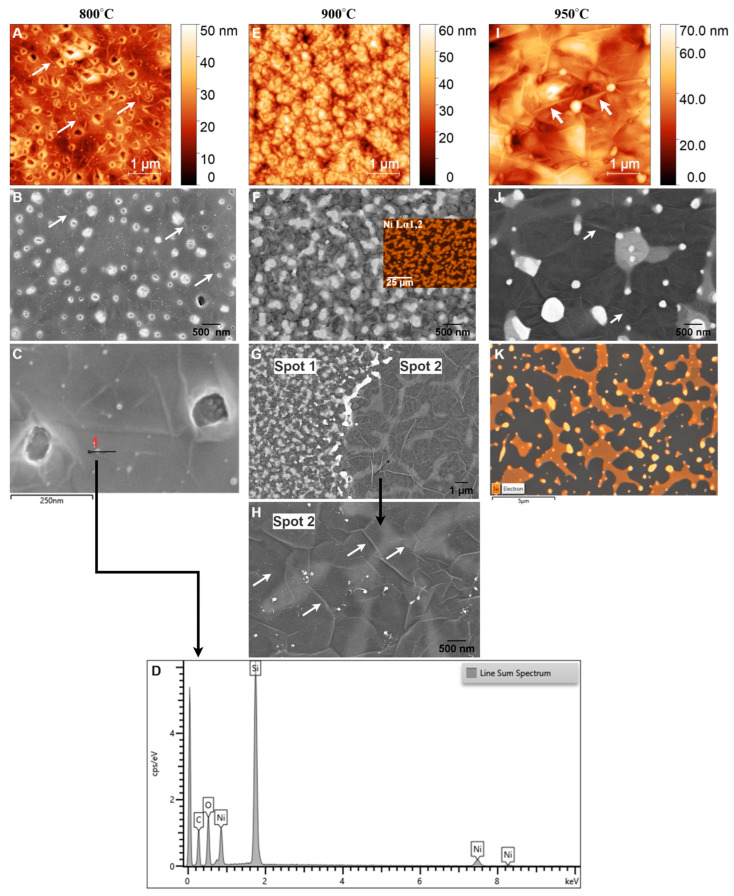
AFM and SEM images of GG films at (**A**–**C**) 800 °C, (**E**–**H**) 900 °C, and (**I**,**J**) 950 °C for 1 min. (**D**) EDX line scan at 800 °C, ((**F**), inset) Ni mapping at 900 °C, and (**K**) Ni mapping at 950 °C. White arrows indicate wrinkles.

**Figure 9 sensors-25-02454-f009:**
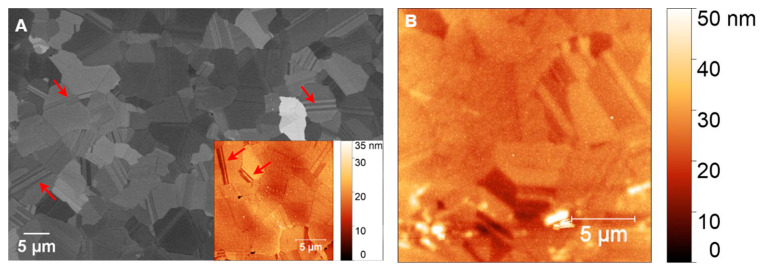
(**A**) SEM image of polished PCD and (**B**) AFM images of nickel-coated glassy carbon on PCD. Arrows indicate twin boundaries. The inset in (**A**) presents an AFM image of polished PCD, highlighting its smooth morphology and fine surface features.

**Figure 10 sensors-25-02454-f010:**
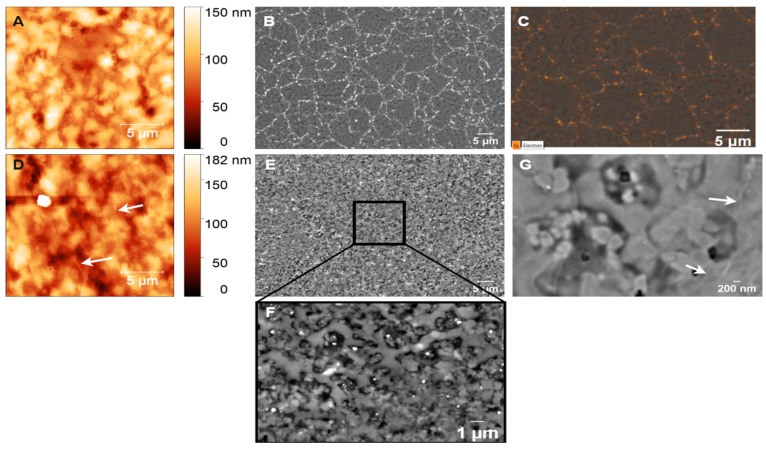
AFM and SEM images of GG on PCD fabricated at 950 °C for (**A**,**B**) 3 s and (**D**–**G**) 1 min. (**C**) EDX mapping of Ni on GG synthesized at 950 °C for 3 s. White arrows indicate wrinkles.

**Figure 11 sensors-25-02454-f011:**
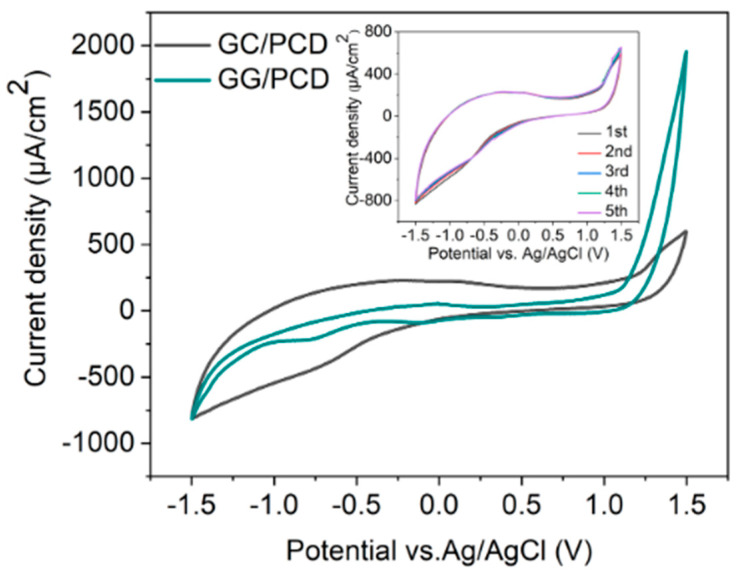
CV measurements recorded in PBS (pH 7.4) at a scan rate of 0.1 V s^−1^ for GC and GG electrodes on PCD, both fabricated at 950 °C for 1 min. The inset displays five consecutive cycles for the GC electrode.

**Figure 12 sensors-25-02454-f012:**
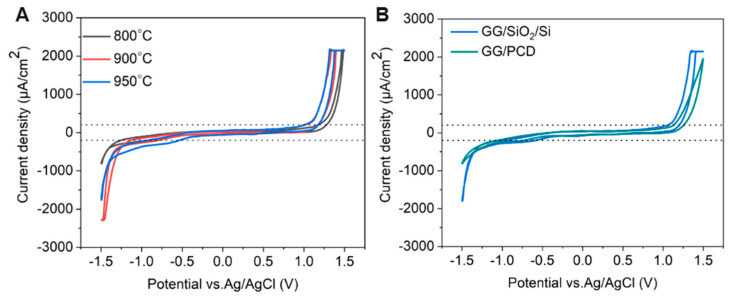
(**A**) CV measurements of GG electrodes on SiO_2_/Si, fabricated at 800, 900, and 950 °C for 1 min in pH 7.4 PBS at a scan rate of 0.1 V s^−1^. (**B**) CV comparison of GG on SiO_2_/Si and PCD, both annealed at 950 °C for 1 min in pH 7.4 PBS at a scan rate of 0.1 V s^−1^.

**Figure 13 sensors-25-02454-f013:**
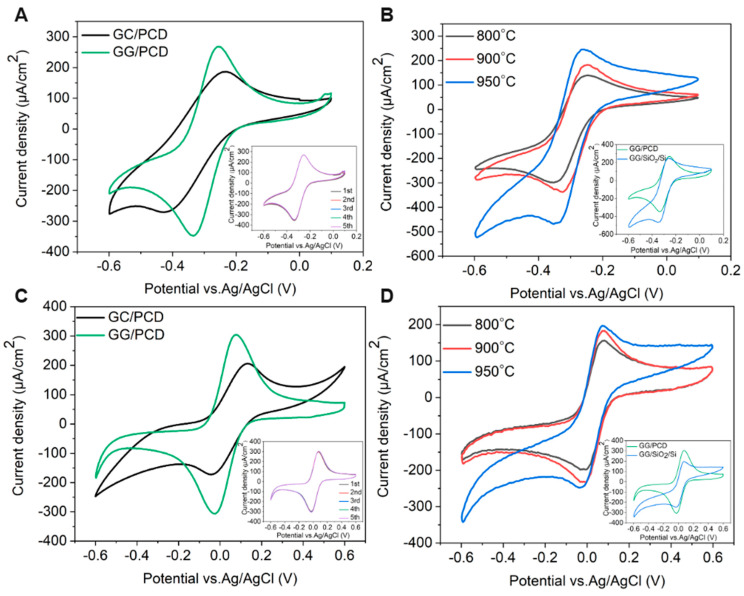
CV measurements of GC and GG electrodes on PCD, both fabricated at 950 °C for 1 min (**A**,**B**) in 1 mM [Ru(NH_3_)_6_]^3+/2+^ and (**C**,**D**) 1 mM [Fe(CN)_6_]^3−/4−^ in 0.1 M KNO_3_ at a scan rate of 0.1 V s^−1^. (**B**,**D**) show CV measurements of GG on a SiO_2_/Si substrate, fabricated at 800, 900, and 950 °C for 1 min. Insets in (**A**,**C**) display five consecutive cycles, while insets in (**B**,**D**) compare CV measurements of GG on SiO_2_/Si and PCD substrates under identical conditions.

**Figure 14 sensors-25-02454-f014:**
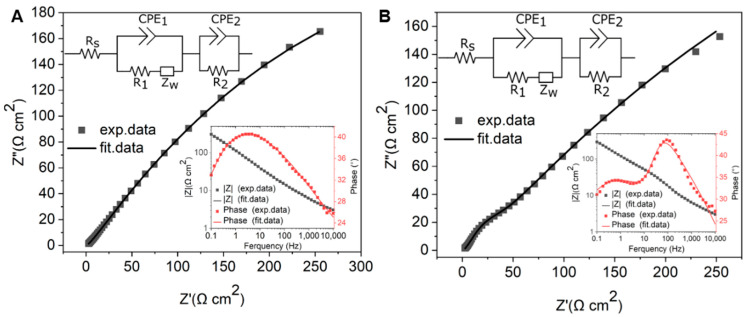
Nyquist plots of GG fabricated at 950 °C for 1 min (**A**) on PCD and (**B**) on SiO_2_/Si in a solution containing 1 mM [Ru(NH_3_)_6_]^3+/2+^ and 0.1 M KNO_3_ at the formal potential. The inset shows Bode plots and equivalent electric circuits.

**Figure 15 sensors-25-02454-f015:**
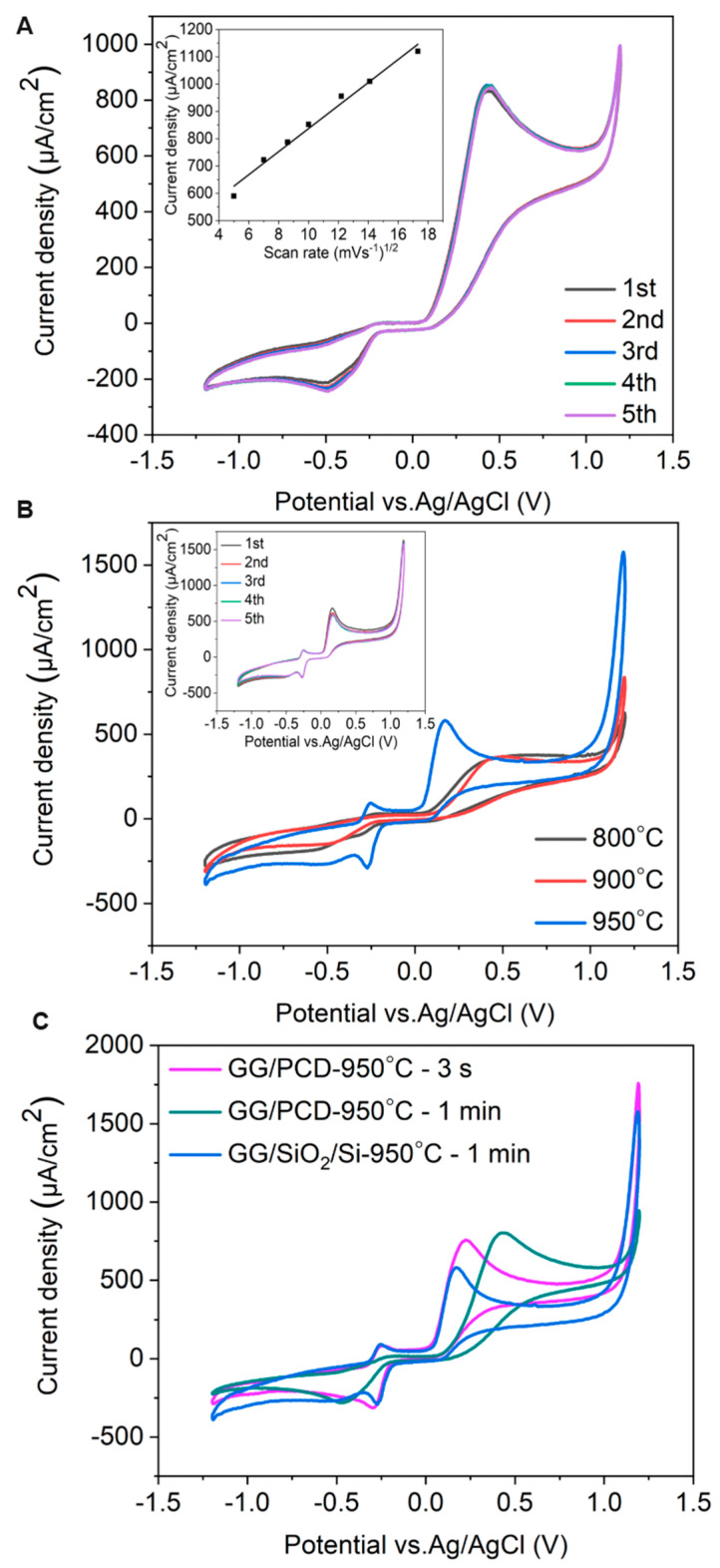
(**A**) Consecutive CV measurements of GG on PCD, fabricated at 950 °C for 1 min. The inset shows oxidation peak currents as functions of the square root of the scan rate (υ^1/2^). (**B**) CV measurements of GG on SiO_2_/Si. (**C**) Comparison of CV measurements of GG on SiO_2_/Si and PCD, fabricated at 950 °C for 3 s and 1 min under identical conditions.

**Figure 16 sensors-25-02454-f016:**
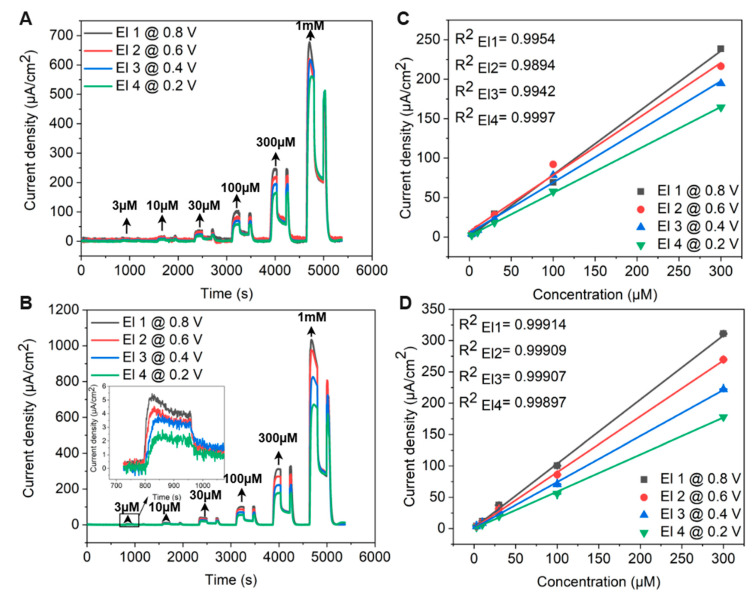
Flow injection analysis with chronoamperometric detection of adrenaline on GG fabricated at 950 °C for (**A**) 3 s and (**B**) 1 min on PCD. Adrenaline concentrations ranged from 3 to 1000 µM in PBS, with each measurement involving a 1.98 mL injection at a flow rate of 0.62 mL/min. The inset displays the response for 3 µM adrenaline. (**C**,**D**) Calibration curves showing the mean current during perfusion on GG chips as a function of applied voltage.

**Figure 17 sensors-25-02454-f017:**
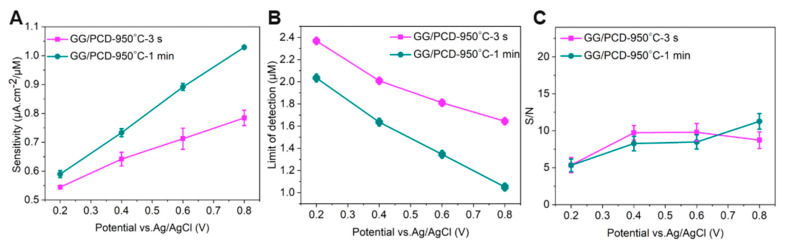
(**A**) Sensitivity, (**B**) limit of detection (LOD), and (**C**) signal-to-noise ratio (S/R) at 3 µM toward adrenaline in flow-injection mode GG/PCD sensors fabricated at 950 °C for 3 s and 1 min.

**Table 1 sensors-25-02454-t001:** The D to G intensity ratio (I_D_/I_G_) and FWHM values of pyrolyzed AZ 5214E photoresist film at 950 °C for 1 min via RTA.

Substrate	I_D_/I_G_	FWHM of D Band (cm^−1^)
SiO_2_/Si	0.43	87.32 ± 1.88
PCD	0.70	102.77 ± 1.86

**Table 2 sensors-25-02454-t002:** Average intensity ratios of D to G (ID/IG) and 2D to G (I2D/IG) of nickel-coated glassy carbon films annealed at 700 °C, 800 °C, 900 °C, and 950 °C for 1 min via RTA.

Substrate	Annealing Temperature (°C)	I_D_/I_G_	I_2D_/I_G_
SiO_2_/Si	700	0.78 ± 0.10	-
	800	0.21 ± 0.04	0.99 ± 0.157
	900	0.17 ± 0.04	0.95 ± 0.277
	950	0.14 ± 0.03	0.80 ± 0.230
PCD	950	0.23 ± 0.01	0.73 ± 0.069

**Table 3 sensors-25-02454-t003:** Electrochemical characteristics of GC and GG electrodes fabricated on SiO_2_/Si and PCD substrates, measured in PBS solution.

Electrode	Annealing Temperature (°C)	Potential Window (V)	Capacitance ^a^ (µF cm^−2^)
GG/SiO_2_/Si	800	2.10	32.48
	900	1.78	33.65
	950	1.49	68.67
GG/PCD	950	1.68	72.98

^a^ Calculated at scan rate of 0.1 V s^−1^.

**Table 4 sensors-25-02454-t004:** Peak-to-peak separation (ΔE_p_) and anodic-to-cathodic peak current density ratio (I_p_^a^/I_p_^c^) for 1mM [Ru(NH_3_)_6_]^3+/2+^ and 1mM [Fe(CN)_6_]^3−/4−^ in 0.1M KNO_3_ on GC and GG electrodes at scan rate of 0.1 V s^−1^.

Electrode	Annealing Temperature (°C)	[Ru(NH_3_)_6_]^3+/2+^		[Fe(CN)_6_]^3−/4−^	
		ΔE_p_ (mV)	I_p_^a^/I_p_^c^	ΔE_p_ (mV)	I_p_^a^/I_p_^c^
GC/PCD	950	163.20 ± 3	0.68	184.00 ± 5	1.20
GG/SiO_2_/Si	800	83.62 ± 1	0.57	103.62 ± 2	0.79
GG/SiO_2_/Si	900	76.96 ± 1	0.55	103.60 ± 2	0.79
GG/SiO_2_/Si	950	90.29 ± 3	0.51	106.96 ± 3	0.80
GG/PCD	950	78.50 ± 2	0.77	98.40 ± 3	0.99

**Table 5 sensors-25-02454-t005:** Comparison of GG/PCD with different electrode materials for adrenaline detection.

Electrode Material	Methods	Linear Range (µM)	Low Detection Limit (µM)	Sensitivity (µA cm^−2^/µM)	References
GR/BDD	CV	1–10	Not reported	1.440	[34]
Caffeic Acid-modified GC	CV	2–300	0.60	Not reported	[79]
LDH-modified GC	DPV	0.5–300	1.0	0.737	[82]
L-Glu, GR-modified GC	DPV	0.1–1000	0.03	0.199	[83]
MXene/N-rGO	DPV	0.01–90	0.003	Not reported	[84]
TiO_2_-rGO	DPV	0.01–0.1	0.0081	0.126	[85]
Porous BDD	FIA	0.6–30	0.50	Not reported	[86]
BDD	FIA	3–1000	0.28	0.920	[42]
GG/PCD	FIA	3–300	1.05	1.029	This work

GR: graphene; BDD: boron-doped diamond; GC: glassy carbon; LDH: layered double hydroxide; L-Glu: L-glutamic acid; N-rGO: N-doped reduced graphene oxide; GG: glassy graphene; PCD: polycrystalline diamond.

## Data Availability

Data can be made available upon written request.

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
