# Peer review of "Novel Amperometric Sensor Based on Glassy Graphene for Flow Injection Analysis"

_sensors, 2025, doi:10.3390/s25082454_

Round 1
Reviewer 1 Report
Comments and Suggestions for Authors
Review of Novel Amperometric Sensor Based on Glassy Graphene for Flow Injection Analysis by Shabgabhi et al.
This paper is very well written and presents a complete fabrication, characterization, and test procedure of a new electrode material for amperometric sensing. The paper starts by describing the fabrication method and its variations, and then presents the material property results in terms of Raman analysis, SEM, AFM, and EDX. The results are promising but not unexpected based on other published work (as acknowledged in the text by the authors). The authors then proceed to perform electrochemical characterization of the electrodes and present CV plots in different supporting solutions and different redox markers. The results are promising. In addition, EIS is performed to understand the charge-transfer dynamics between the electrode and the solution. Then, CV plots are obtained for adrenaline-containing solutions. Finally, microfluidic detection of adrenaline is done within the favorable conditions determined above.
The authors are presenting a lot of characterization data and discussing these data in detail, which is appreciated. The authors should consider moving some of this data to a supplementary file and providing the results on the main text so that the reader remains engage throughout the manuscript.
The introduction of the paper focuses on the fabrication of graphene-based electrodes, some of the attempts published in literature, and their results as sensing in different chemicals. It also introduces and describes the method of flow injection analysis as opposed to a diffusion method and it states some of its advantages.
Nevertheless, there is a slight disconnect between this introduction and the flow of the paper. The paper starts with the fabrication process and does a very thorough job analyzing the physical results as a response to the different process variations. Then it goes on to spend quite a bit of time characterizing the electrochemical behavior. As such, it would be helpful to include in the introduction some of these electrochemical characterization techniques, why they are important, and applicable published results as they relate to this type of electrode and sensing applications.
Why talk about dopamine and catecholamines detection in lines 59, 60? A different paragraph and a transition sentence may be needed in the introduction to describe some of the common chemicals sensed with these types of electrodes and why this study focuses on adrenaline.
What is the starting Raman of AZ 514E? Any ordered domains?
There is a big difference between locations in the Raman peak ratios. Can you expand on the non-uniformity of the nickel catalyst? With this much variation, what is the statistical significance of the reported values of Table 2 and the related discussion? (Only 3 locations on one sample? Or is this typically observed across many samples?
Subscripts for H2O and CO2 in lines 411 and 412.
Only the GG/PCD, 950C for 3 s and 1 min were used as sensors; as such, some of the conclusions that the authors arrived at with respect to the CV analysis results cannot be confirmed in the final application. Would the authors consider fabricating additional SiO2/Si electrodes to confirm the lower detection capabilities of these as opposed to those made on the PCD substrates?
In addition, how do the presented sensing results compare to other adrenaline sensors based on different electrodes and/or flow conditions? A comparison discussion or a table would be helpful.
Reviewer 2 Report
Comments and Suggestions for Authors
Reviewers' comments:
Reviewer #1 (Remarks to the Author):
The manuscript entitled “Novel Amperometric Sensor Based on Glassy Graphene for Flow Injection Analysis” by Shabgahi and co-workers describes the electrochemical sensing performance of glassy graphene electrodes derived from pyrolyzed positive photoresist films (PPFs) via rapid thermal annealing (RTA) on SiO2/Si and polycrystalline diamond (PCD). Glassy graphene films fabricated at 800, 900, and 950 °C were characterized using Raman spectroscopy, scanning electron microscopy (SEM), and atomic force microscopy (AFM) to assess their structural and morphological properties. Electrochemical characterization in phosphate-buffered saline (PBS, pH 7.4) revealed that annealing temperature and substrate type influence the potential window and double-layer capacitance. FIA with amperometric detection showed a linear electrochemical response to adrenaline in the 3-300 µM range, achieving a low detection limit of 1.05 µM and high sensitivity of 1.02 µA cm-2/µM.
It was a pleasure for me reading through this manuscript.
However:
To my opinion, unfortunately, some minor corrections are required to make the manuscript more attractive.
Minor comments:
- The author should have included a schematic illustration describing the experiment and findings below the introduction in the manuscript. So that it will be clear to the readers as there are multiple detections involved.
- In line 315 “The increased ID/IG ratio and D-band broadening…” should be replaced as “The increased ID/IG ratio ……”. Please correct this sentence.
- In line 332-333 and 365 “Additionally, the ID/IG ratio is higher compared….” Should be replaced as “Additionally, the ID/IG ratio…….”.
- In line 350 and 353 “the I2D/IG…..” should be replaced to “I2D/IG”
- In line 411-412 “compounds such as H2O, CO, and CO2..” should be replaced as “compounds such as H2O, CO, and CO2….”
- In line 182, the author has written a sentence regarding the Raman Spectroscopy that “Spectra were acquired using a 633 nm laser….”. After going thoroughly in 3.1 section, I never came across the 532 nm laser analysis from the author. The author should clearly state regarding why he has not performed this experiment as this is commonly considered as one of the best choices because of its strong resonance with graphene and its good signal intensity?
- The author stating “The peak at 1593 cm-1, known as the G band, is attributed to the in-plane vibrational modes of sp²-bonded carbon atoms, a feature also observed in graphite. This band is indicative of graphitic domains and serves as a marker of sp² carbon ordering within the material” should be properly cited with reference like (https://doi.org/10.1016/j.mser.2024.100805).
After these minor corrections, the manuscript is well suited to publish in mdpi Sensors Journal.

Reviewer 3 Report
Comments and Suggestions for Authors
In this manuscript, Shabgahi et al. introduce a novel approach for the fabrication of glassy graphene-based sensors on polycrystalline diamond and SiO2/Si substrates by rapid thermal annealing processes. Their data are consistent and well presented. The discussion of the scientific results is fluid and engaging to read. So, this reviewer thinks this manuscript will be publishable after a few minor revisions.
Materials and Methods
- What kind of tip was used for AFM measurements? Could the authors report this information in the materials and methods section.
- In line 183 the authors report that the laser power used for Raman measurements is 5% of its maximum power. How much does this value correspond to? The authors should report the value of the power in watts.
- Regarding the fabrication of glassy graphene (GG) microelectrodes, in line 271 the authors report that sonication was used to dissolve the remaining photoresist and remove the excess Ni. The authors should quantify how long the sample was sonicated.
- Figure 2 which shows the workflow of the various experimental steps is well designed, clear and articulated. However, it should be described and commented in more detail either in the text or in the caption of the figure.
Results
- In paragraph 3, there is also a discussion of the results. Therefore, this reviewer suggests that it would be more appropriate to title it “Results and discussions”.
- In line 289 the authors introduce their discussion of Raman spectroscopy as a powerful tool to study and characterize the transformation of samples during the various processes. The discussion is well structured, clear and comprehensive. However, if this reviewer understands correctly, all the presented spectra are fitted curves derived from the experimental data. It would be helpful to explain why the fits are shown and not the experimental curves. What is the signal to noise ratio? The authors could add the raw Raman data in the supplementary material, reporting the intensity values on the y-axis.
- In Figure 3 the authors report a Raman spectrum of the PPF on SiO2/Si (A) and on PCD (B). The spectra are commented in detail and the differences between the two are well underlined. Was a statistical investigation carried out on different points of the sample to confirm the intensity ratios of the bands reported in Table 1?
- How was calculated the experimental error of the FWHM values of D Band in Table 1?
- The material characterization is complete and rigorous in all its parts. However, the characterization of the used electrodes (after electrochemical tests) is absent. Could the authors comment on this? According to this reviewer, a morphological investigation of the modified electrodes could better explain the sensor recyclability properties.
- A table with a comparison of the performances of similar materials from other works could enhance the quality of the manuscript.
- Please check the superscripts and subscripts throughout the manuscript (mainly in paragraph 3.1.2.).

Reviewer 4 Report
Comments and Suggestions for Authors
In this work (sensors-3553934), the authors describe the preparation of glassy graphene electrodes, as well as the evaluation of their structural, morphological, and electrochemical properties. Subsequently, the combination of the glassy graphene electrode and FIA was used for electrochemical sensing of adrenaline.
Since the development of new electrode materials is a highly demanded area of research in electroanalysis, the results of this research work can be considered scientifically significant and impactful. The research topic is fully in line with the scope of ‘Sensors’ journal. The work is written in good English. Nevertheless, the paper contains some points that need to be improved (see the list below).
- The authors introduced the abbreviations ‘GC’ and ‘GG’ but continued to use the full names, glassy carbon and glassy graphene, respectively.
- The abbreviation ‘PBS’ is introduced several times (lines 165, 218, 618, 881).
- Lines 620-622: why were the anodic and cathodic limits identified as the potential where current density exceeded ±200 µA/cm2, while in the previous paper on the BDDE, cited as [38] in this manuscript, the density of ±15 µA/cm2 was used? It is clear that the larger the current density is used to define the anodic/cathodic limit, the wider the potential window is obtained. Please explain.
- Table 3: for what scan rate was the capacitance calculated?
- It is stated that ‘Upon the conversion of glassy carbon to glassy graphene, a significant reduction in double-layer capacitance is observed (Figure 11) and summarized in Table 3’ (lines 646-647). However, the reduction in capacitance cannot be estimated because the capacitance for glassy carbon is not listed in Table 3 or in the text.
- The caption to Figure 11 does not describe the inset.
- The authors discuss the change in the electrochemically active surface area between the glassy graphene electrode and the glassy carbon electrode (lines 639, 667, 778, 817). However, this discussion is not supported by the data – I have not found the calculated values of the electrochemically active surface area of the electrodes.
- The CVs presented in Figures 12 and 13 could be combined and presented in one figure.
- Table 4: results should be presented with the same number of significant figures (significant digits).
- Section 3.3.2: to evaluate the electrode quality, in addition to the anodic and cathodic peak potential difference (∆E), the anodic peak current-to-cathodic peak current ratio (Ia/Ic) of the redox markers should also be taken into account. These values could be added to Table 4.
- Error bars should be added to Figures 17B and 17C.
Round 2
Reviewer 1 Report
Comments and Suggestions for Authors
The authors have addressed this reviewer's comments very well. Thank you for a great paper!
Reviewer 4 Report
Comments and Suggestions for Authors
Manuscript: sensors-3553934
The authors have answered all the questions and revised the manuscript to address all the comments. Consequently, the manuscript has been significantly improved to warrant publication in Sensors.
Brief comments on the authors’ responses #7 and #11:
(i) electroactive surface area of the electrodes can be easily calculated using the Randles-Sevcik equation from the CV data of the redox marker, ([Ru(NH3)6]3+/2+ or [Fe(CN)6]3−/4− ;
(ii) if the error bars are very low and are not visible due to their overlapping with the symbol, in Origin it can be easily fixed by changing the colors of error bar to a different color than the symbol and selecting ‘through symbol’ as the placement of the error bars.